# Meltwater runoff and glacier mass balance in the high Arctic: 1991-2022 simulations for Svalbard

Louise Steffensen Schmidt[1], Thomas V. Schuler[1], Erin Emily Thomas[2,3], and Sebastian Westermann[1]

[1]Department of Geosciences, University of Oslo, Norway
[2]Norwegian Meteorological Institute, Oslo, Norway.
[3]Current Affiliation: Fluid Dynamics and Solid Mechanics, Los Alamos National Laboratory, Los Alamos, NM, USA

**Correspondence:** Louise Steffensen Schmidt (l.s.schmidt@geo.uio.no)

**Abstract.** The Arctic is undergoing increased warming compared to the global mean, which has major implications for freshwater runoff into the oceans from seasonal snow and glaciers. Here, we present high-resolution (2.5 km) simulations of glacier mass balance, runoff and snow conditions in Svalbard from 1991-2022, one of the fastest warming regions in the world. The simulations are created using the CryoGrid community model forced by CARRA reanalysis (1991-2021) and AROME-ARCTIC forecasts (2016-2022). Updates to the water percolation and runoff schemes are implemented in the CryoGrid model for the simulations. In-situ observations available for Svalbard, including automatic weather station data, stake measurements, and discharge observations, are used to carefully evaluate the quality of the simulations and model forcing.

We find a slightly negative climatic mass balance (CMB) over the simulation period of -0.08 m w.e. $\mathrm{yr}^{-1}$, but with no statistically significant trend. The most negative annual CMB is found for Nordenskiöldland (-0.73m w.e. $\mathrm{yr}^{-1}$), with a significant negative trend of -0.27m w.e. $\mathrm{decade}^{-1}$ for the region. Although there is no trend in the annual CMB, we do find a significant increasing trend in the runoff from glaciers of 0.14 m w.e. $\mathrm{decade}^{-1}$. The average runoff was found to be 0.8 m w.e. $\mathrm{yr}^{-1}$. We also find a significant negative trend in the refreezing of -0.13m w.e. $\mathrm{decade}^{-1}$.

Using AROME-ARCTIC forcing, we find that 2021/22 has the most negative CMB and highest runoff over the 1991-2022 simulation period investigated in this study. We find the simulated climatic mass balance and runoff using CARRA and AROME-ARCTIC forcing are similar, and differ by only 0.1 m w.e. $\mathrm{yr}^{-1}$ in climatic mass balance and by 0.2 m w.e. $\mathrm{yr}^{-1}$ in glacier runoff when averaged over all of Svalbard. There is, however, a clear difference over Nordenskiöldland, where AROME-ARCTIC simulates significantly higher mass balance and significantly lower runoff. This indicates that AROME-ARCTIC may provide similar high-quality predictions of the total mass balance of Svalbard as CARRA, but regional uncertainties should be taken into consideration.

The simulations produced for this study are made publicly available at a daily and monthly resolution, and these high resolution simulation may be re-used in a wide range of applications including studies on glacial runoff, ocean currents, and ecosystems.

# 1 Introduction

Glaciers and ice caps are considered to be good indicators of climate change. During the last decades, glaciers and ice caps worldwide have been responding to a globally warming climate by melting at increasing rates (e.g. Vaughan et al., 2013; Huss and Hock, 2018). The Arctic has experienced greater warming than the global average due to positive feedbacks triggered by changing sea ice cover, the so-called Arctic amplification (e.g. Serreze and Francis, 2006; Graversen et al., 2008; Lind et al., 2018). As sea ice continues to retreat, further warming in the Arctic is expected (e.g. IPCC, 2019).

In particular, the region around the Barents Sea, which includes the archipelagos of Svalbard, Franz Josef Land and Novaya Zemlya, has experienced pronounced warming in recent decades due to disappearing sea ice (e.g. Screen and Simmonds, 2010; Lind et al., 2018; Isaksen et al., 2022). For example, the Svalbard archipelago has the strongest observed warming in Europe since the 1960's, with temperatures increasing at a rate of $0.5°\mathrm{C}\,\mathrm{decade}^{-1}$ (Nordli et al., 2014). Even under the moderate RCP4.5 emission scenario, which projects a global temperature increase of $1.1 - 2.6°\mathrm{C}$ by 2100 relative to the 1986-2005 period, temperatures in the Barents Sea region are projected to increase by 5-9°C (AMAP, 2017; Hanssen-Bauer et al., 2019).

Although the volumes of ice on Svalbard is only equivalent to a global sea level rise of about 15 mm (Fürst et al., 2018), it is estimated to be one of the most important regional contributors to sea level rise in the 21st century (e.g. Meier et al., 2007; Church et al., 2013; Hock et al., 2019). In addition to sea level rise, meltwater from retreating glaciers is important for river hydrology, fjord circulation, and terrestrial and marine ecosystems (e.g. Carroll et al., 2017; Hopwood et al., 2020).

Observations of meltwater runoff from glaciers in Svalbard is challenging, and only a couple of partially glaciated catchments are continuously monitored (Sund and Monica, 2008). However, glaciological measurements of the surface mass balance (SMB) have been conducted on Svalbard since the 1960's (e.g. Hagen et al., 2003; Schuler et al., 2020), but these observations also only exist in a small area. Therefore, dedicated energy/mass balance models are an important tool to determine the runoff and mass balance of the whole archipelago.

To simulate the runoff and mass balance of the past using a physically based energy balance model, it is important to have accurate estimates of the meteorological forcing (temperature, wind speed, humidity, incoming radiation, and precipitation). Global reanalysis products, such as ERA-Interim (Dee et al., 2011) and ERA5 (Hersbach et al., 2020), provide reliable estimates of the past atmospheric conditions, but the resolution of these products is too coarse to properly resolve ice caps and glaciers in the Arctic. Previous studies of the mass balance of Svalbard have further downscaled these global products either using a regional climate model (e.g. Lang et al., 2015; Aas et al., 2016), statistical downscaling (e.g. Østby et al., 2017), or a combination of both (e.g. Van Pelt et al., 2019). However, statistical downscaling does not fully resolve the physical processes of the atmosphere, and thus may introduce further uncertainties, while regional climate models are computationally expensive.

Regional reanalysis products, which provide a physical downscaling of global reanalysis while assimilating additional global simulations, may be the best solution to this problem. In recent years, high-resolution simulations of the meteorological conditions over the Arctic have become available. In late 2015, forecast simulations from the high-resolution (2.5 x 2.5 km) AROME-ARCTIC numerical weather prediction system became available over the Barents Sea region, based on the state-of-the-art numerical weather simulation model HARMONIE-AROME (Bengtsson et al., 2017). This system assimilated available

regional observations. It has been used as forcing for different short-term climate studies on Svalbard (e.g. Zweigel et al., 2021; Schmidt et al., 2021). In 2021, the high-resolution Copernicus Arctic Regional ReAnalysis (CARRA) dataset (Schyberg et al., 2020; Yang et al., 2021) was published. It is a reanalysis product with a 2.5 km resolution, downscaled from ERA5 (Hersbach et al., 2020) by the state-of-the-art weather prediction model HARMONIE-AROME (Bengtsson et al., 2017). CARRA includes a number of improvements over ERA5, such as the assimilation of a large amount of additional surface observations, extensive use of satellite data, and an improved representation of sea ice. CARRA is likely the best high-resolution estimate of the meteorological parameters available in the Barents Sea region currently available due to the complex physics contained within the model and the large amount of assimilated data. It has been shown that CARRA has improved general verification statistics for all simulated regions compared to ERA5, with the largest differences assosiated with complex terrain (Køltzow et al., 2022).Further downscaling is not required since it already contains such a high spatial resolution, which avoids introducing more uncertainties.

Here, we evaluate the use of the novel CARRA product for simulations of the mass balance and runoff of Svalbard from 1991-2021, and investigate the changes in these parameters over the last three decades. The forcing is thoroughly evaluated against observations, to assess the uncertainties of the product over glaciers. In addition, we investigate if the forecast product AROME-ARCTIC, which uses the same model and similar observations as CARRA, can be used to extend the CARRA product, thus providing almost real-time updates of the mass balance and runoff. Almost real time-simulations could provide valuable information for e.g. fieldwork planning (to check the current conditions) and public outreach. AROME-ARCTIC is also evaluated against observation, and we compare the results from the two products for the period they overlap. Although other products based on HARMONIE-AROME have successfully been used as forcing for mass balance simulations in the Arctic (e.g. Mottram et al., 2017; Schmidt et al., 2018), neither AROME-ARCTIC nor CARRA have previously been validated for use in mass balance simulations.

The mass balance simulations are conducted using CryoGrid, a physical-based model for simulating the terrestrial cryosphere (Westermann et al., 2023). CryoGrid can be applied to a large range of Arctic environments, and simulates the energy and mass balance of both seasonal snow and glaciers, and estimates permafrost in non-glaciated areas. The CryoGrid model results are evaluated against available observations of mass balance, both from in-situ campaigns and geodetic methods. CryoGrid simulates both the surface mass balance (SMB) and the climatic mass balance (CMB). The SMB quantifies the mass fluxes between the atmosphere and the glacier at the surface, as well as refreezing within the annual layer. The SMB is what is measured by in-situ glaciological observations. The CMB additionally accounts for mass changes below the last summer surface, e.g. in the deeper firn layers. The total mass balance, the sum of CMB, basal mass balance and frontal ablation, cannot be calculated for tidewater glaciers by an energy-balance model like CryoGrid, as glacier dynamics are not included. This terminology follows that suggested by Cogley et al. (2011).

As a result of this study, the surface/climatic mass balance, runoff, refreezing, and seasonal snow amount of Svalbard from 1991-2022 is presented and evaluated. We provides an update on the mass balance of Svalbard compared to previous studies,

and look at the current trends in the mass balance components and runoff. The produced simulations are provided with the paper, and may be used for a wide range of future applications, e.g. as input for runoff, ocean circulation, or ecosystem models.

## 2 Study area

Located in the Norwegian Arctic between 75 and 81°N, the Svalbard archipelago is in one of the currently fastest warming regions in the world, the Barents Sea region (e.g. Screen and Simmonds, 2010; Lind et al., 2018), and has the strongest observed warming in Europe since the 1960's (Nordli et al., 2014). With a land area of $\sim$60,000 $km^2$, of which about 57% is covered by glaciers (Nuth et al., 2013), it contains about 10% of the glacier area in the Arctic, outside of the Greenland ice sheet. The glacier types vary between small cirque and valley glaciers to large ice fields and ice caps, with more than 1000 individual mapped glaciers across the archipelago. Around 60% of the glacier area belong to tidewater glaciers (Błaszczyk et al., 2009) which introduce freshwater into the oceans through discharge from subglacial channels or calving at the glacier front. The highest elevations on Svalbard reach 1700 m a.s.l., but the hypsometry of glaciers peaks at 450 m a.s.l (Noël et al., 2020).

While the western side of the Archipelago is kept warm and humid by the Norwegian current, which brings warm Atlantic currents northwards along the western coast (Walczowski and Piechura, 2011), and warm and moist air from southerly flows, the eastern side is colder and drier, dominated by the cold Arctic ocean current and dry and moist air masses originating in the north-east (Käsmacher and Schneider, 2011). Precipitation varies wildly across the archipelago, with the highest precipitation rates in the south and along the west coast (Førland and Hanssen-Bauer, 2003; Winther et al., 2003; Førland et al., 2020). These patterns in temperature and moisture are reflected in the distribution of glaciers, with the largest glaciers found in the north-east and less extensive glacier coverage along the western side of Svalbard and in central Spitsbergen.

## 3 Methods and Data

### 3.1 Methods

The simulations presented in this paper were created using the full energy balance model CryoGrid, which is forced by both CARRA reanalysis and AROME-ARCTIC forecasts. The workflow used is described below.

First, both the CARRA reanalysis and AROME-ARCTIC forecasts are evaluated against available observations from automatic weather stations (AWSs). Unsurprisingly, both products performed well when compared to AWSs which had been assimilated into the forcing products, but had larger biases when compared to glacier AWSs which had not been assimilated. The comparison of AROME-ARCTIC and CARRA at the AWS locations were similar, albeit with larger biases and root-mean-square-errors for AROME-ARCTIC. In addition, the consistency between the two forcings is evaluated for the overlap period (2016-2021). We found that AROME-ARCTIC is on average colder than CARRA, particularly in NW Spitsbergen where the average yearly temperature was -2°C colder in AROME-ARCTIC. The full results of this analysis are described in Supplement S2.

We then perform a 30-year spin-up of the CryoGrid model (described in section 4) for the glaciated gridpoints by repeating the 1991-2000 CARRA forcing to initialise the snow/ice temperature, density, and water content. The model is initialised with 47 layers of ice with a thickness between 0.1 and 1 m, totalling 20 m of glacier ice. Initially, the entire domain consists of temperate, pure glacier ice, i.e. the ice temperature of the entire column is 0°C. Tests were conducted with lower initial temperatures (-5°C), but it did not affect the temperature profile at the end of the spin-up. At the end of the spin-up period, the runoff, refreezing, subsurface temperatures and climatic mass balance reached stable values. For the non-glaciated land points, only a 2-year spin-up was used.

The energy/mass balance model CryoGrid is then used to simulate the mass balance components of both glaciers and seasonal snow from 1991-2021 using the CARRA reanalysis as forcing. The output from the CryoGrid simulations is evaluated against in-situ mass balance observations and geodetic estimates. More details on the evaluation is provided later in this section.

Lastly, a second simulation with CryoGrid, this time forced by AROME-ARCTIC, is conducted from 2016 to present. From 2016 until the summer of 2019, the AROME-ARCTIC model was initialized with too little snow over some glacier points in the ablation area, thus leading to unrealistically high surface and 2m temperatures. To counter this effect, we use the 10m temperature for the AROME-ARCTIC-forced simulation when unrealistically high surface temperatures occur. The AROME-ARCTIC-forced simulation is initialized from the CARRA-forced simulation at the end of 2015. Thus, the initial conditions for the 2016-2021 period are identical for the two simulations. This most likely will reduce the difference in CMB calculated using the two products, compared to if different spin-ups were produced. The AROME-ARCTIC-forced CryoGrid simulation are currently automatically updated on a daily basis. In this study, we present the simulations spanning until September 1st, 2022.

For the CryoGrid simulations, a fractional glacier mask is created by computing the percentage of glacier coverage in each grid point. The glacier coverage is based on the extent in the 2000s, based on the inventory of Nuth et al. (2013). Any points which have a fractional coverage between 10 and 90% are calculated with both the glaciated and non-glaciated land schemes. To calculate the average or sum of a variable for a specific region or all of Svalbard, the results are weighted based on the fractional glacier coverage.

## 3.2 CryoGrid Model Forcing

Meteorological forcing fields of 2m air temperature, specific humidity, incoming long- and shortwave radiation, pressure, and mass fluxes were obtained from both the Copernicus Arctic Regional ReAnalysis (CARRA) dataset (Schyberg et al., 2020; Yang et al., 2021) and AROME-ARCTIC weather forecasts (e.g. Müller et al., 2017).

CARRA is based on the non-hydrostatic numerical weather prediction model HARMONIE-AROME (Bengtsson et al., 2017). It uses ERA5 reanalysis (Hersbach et al., 2020) as boundary conditions and downscales it to a 2.5 x 2.5 km resolution over the European Arctic. The simulations are divided into two domains. Here we use the east domain, which contains Svalbard, Franz Josef Land, Novaya Zemlya and Northern Norway. CARRA currently spans the time frame from September 1990 to December 2021.

Similar to CARRA, AROME-ARCTIC (e.g. Müller et al., 2017) is also based on HARMONIE-AROME, and provides operational forecasts at a 2.5 x 2.5 km resolution over the Barents sea region. It uses ECMWF HRES forecasts as a lateral boundary conditions. The model has been operated by the Norwegian Meteorological Office since October 2015 and provides 66-hour forecasts with hourly resolution every 6-hours. Since this is a real-time forecast product, there are occasionally gaps in the forecast. When possible, we use the forecast initialized at 18UTC, as most data is assimilated at this time. We use a

6-hour lead time and extract data for 24 hours at a time, thus using forecast timesteps 6-30 for the simulations. This is chosen to optimise the prediction quality as well as to avoid spin-up effects. When the 18UTC forecast is not available, we use longer lead times of previous forecasts to find the most recent available estimate at a given hour. In the rare case that no forecast is available for the desired period, we simply interpolate between the previous and following available timestep.

    Since CARRA and AROME-ARCTIC are on slightly different grids, we bilinearly interpolate the AROME-ARCTIC fields

onto the CARRA grid in order for the final dataset to be consistent.

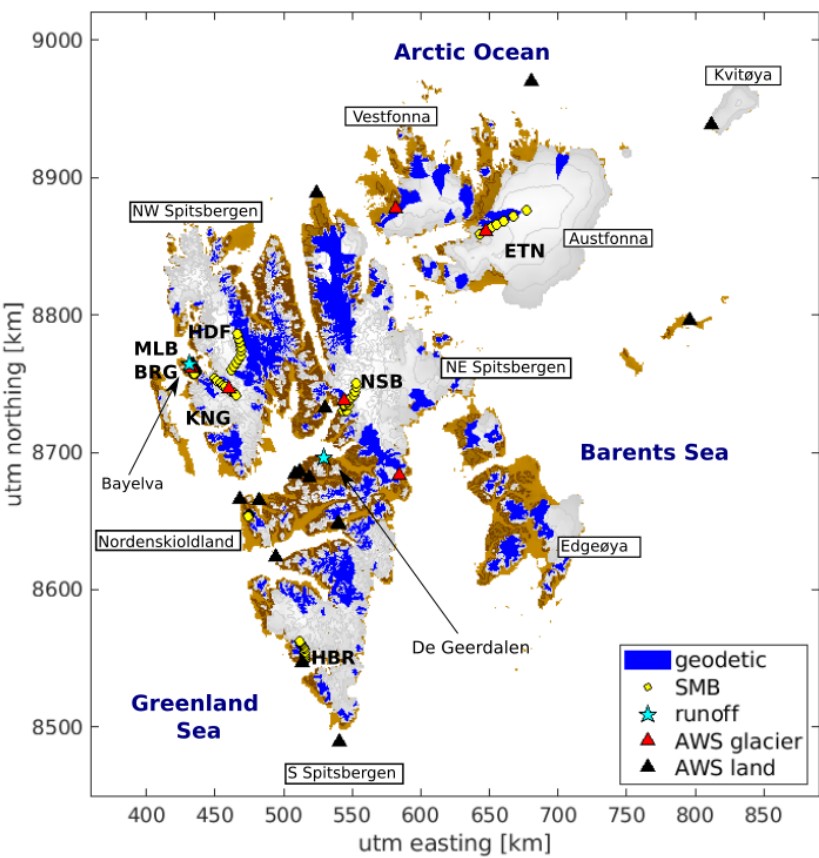

**Figure 1.** The location of the surface mass balance stakes and automatic weather stations used (Table 1) and the names of the different regions. The blue shaded areas are used for comparison with geodetic mass balance estimates.

### 3.3 In-situ data

For evaluation of the model forcing, observations from 26 Automatic Weather Stations (AWSs) are used: 6 stations on glaciers and 20 stations on non-glaciated land (see Fig. 1 and Table 1). The 20 stations on non-glaciated land are all operated by the Meteorological Office in Norway (MET-Norway) and have been assimilated into the CARRA and AROME-ARCTIC products. The 6 glacier stations, on the other hand, have not been assimilated and thus provide independent reference. The glacier stations are located on: Etonbreen, operated by University of Oslo and the Norwegian Polar institute since 2004, (e.g. Schuler et al., 2014), Kongsvegen, operated since 2007 by the Norwegian Polar Institute (Kohler et al., 2017), Vestfonna, operated for two years between 2007-2009 by Uppsala University (Jonsell, 2017), and Nordenskiöldbreen and Ulvebreen, operated by Utrecht University since 2009 and 2015, respectively. The measurement interval was between 1-2 minutes, depending on the station. The measurement height varies between stations and during the year due to accumulation of snow below the sensors. When available, daily mean observations of the 2m temperature, 2m relative humidity, 10m wind speed, and incoming and outgoing longwave and shortwave radiation is used for the evaluation. When windspeed is only available below 10 m, as is the case for most of the glacier stations, the windspeed at 10 m is calculated using a logarithmic wind profile (assuming neutral stratification) with a roughness length of 1 mm. The assumption of neutral stratification, however, is a limitation, potentially having larger impact on the wind speed correction than sensor level alone. For Nordenskiöldbreen and Ulvebreen, measurements were conducted at approx 4m above the surface, and the CARRA humidity and temperature is therefore interpolated to the measurement height by interpolation between the lowest model level (15 m) and 2 m.

The snowdepth is not measured at the majority of the used stations, and we therefore do not apply any correction factor due to changes in height after snow accumulation. The uncertainty associated with ignoring this effect depends on the specific variable (temperature, humidity, windspeed) and the measurement height. These uncertainties only affect the evaluation statistics, and not the model results.

Snow depths on Svalbard are modest and seldom amount to more than 1 m at most AWS sites. Assuming a snow depth of 1 m, a roughness length of snow of 1 mm, and that the windspeed can be approximated by a logarithmic profile (neutral stratification), the windspeed at 1 m above the surface is 7% lower than the windspeed at 2 m. For windspeeds measured at 10 m, decreasing the height by 1 m only amounts to a 1% decrease in windspeed. The windspeeds measured at the MET Norway stations and Kongsvegen, which are measured at 10 m, are therefore more robust to the effect of snow accumulation. The study by Østby et al. (2013) suggests a roughness length smaller than 1 mm which in turn would decrease the effect on wind speed.

It is trickier to estimate the uncertainties for temperature and relative humidity. Here, we use CARRA estimates of the temperature, pressure, wind speed, and humidity at the lowest model level (15 m) and at surface level to interpolate the temperature and specific humidity, taking into account the stability of the atmosphere. The same method and parameters are used within CARRA to calculate variables at 2m height, and is described in detail in the CARRA product user guide (Schyberg et al., 2020). The difference in temperature and humidity for all station locations is simulated for 2 m and 1 m above the surface over two different years (1994, a low melt year, and 2020, a high melt year). Even assuming that the snowpack lasted the full year, the yearly average deviation was $< 0.2°C$. The specific humidity at surface level was not available in CARRA, so for simplicity

we assume fully saturated conditions. The yearly average difference in the results was always below 1%.

**Table 1.** In-situ data used for evaluation of the forcing, surface mass balance, and runoff. The locations of the measurement points are shown in Fig. 1. UiO: University of Oslo, NPI: Norwegian Polar Institute, IMAU: Institute for Marine and Atmospheric research Utrecht, PAN: Polish Academy of Sciences, UU: Uppsala University, NVE: The Norwegian Water Resources and Energy Directorate.

| Description | Location | Elevation | Period used | Frequency | Source |
|---|---|---|---|---|---|
| Automatic Weather Stations | Etonbreen (22.4E, 79.7N) | 370 m a.s.l. | 2004 – 2020 | daily | UiO |
| | Kongsvegen (78.8E, 13.2N) | 537 m a.s.l. | 2007 – 2016 | daily | NPI |
| | Nordenskiöldbreen (78.7E, 17.0N) | 530 m a.s.l. | 2009 – 2020 | daily | IMAU |
| | Ulvebreen (78.2E, 18.7N) | 140 m a.s.l. | 2015 – 2020 | daily | IMAU |
| | Vestfonna (78.8E, 13.2N) | 305 m a.s.l. | 2007 – 2009 | daily | UU |
| Mass balance stakes | Austre Brøggerbreen (BRG) | | 1991 – 2018 | summer, winter | NPI |
| | Midtre Loveenbreen (MLB) | | 1991 – 2018 | summer, winter | NPI |
| | Kongsvegen (KNG) | | 1991 – 2018 | summer, winter | NPI |
| | Hansbreen (HBR) | | 1991 – 2012 | summer, winter | PAN |
| | Holtedahlfonna (HDF) | | 2003 – 2018 | summer, winter | NPI |
| | Linnébreen (LNB) | | 2004 – 2010 | summer, winter | NPI |
| | Etonbreen (ETN) | | 2004 – 2018 | summer, winter | UiO, NPI |
| | Nordenskiöldbreen (NSB) | | 2006 – 2018 | summer, winter | IMAU, UU |
| Discharge observations | Bayelva (78.9E, 11.8N) | | 1991 – 2022 | daily | NVE |
| | De Geerdalen (78.3E, 16.3N) | | 1991 – 2022 | daily | NVE |

In addition, observations from mass balance stakes are used for evaluation of the CryoGrid products. When several observation points fall within one 2.5 x 2.5 km model grid, only the measurement point closest to the center of the gridpoint is used. A total of 52 measurement points are used, spread over 8 glaciers and ice caps (Table 1 and Fig. 1). The stake heights
are recorded once or twice a year (typically in April and September), and are converted into summer and winter mass balance estimates using snow density and snow depth data. Stake data on Austre Brøggerbreen (BRG), Midtre Lovénbreen (MLB), Kongsvegen (KNG), Holtedahlfonna (HDF), and Linnébreen (LNB) have been collected by the Norwegian Polar Institute (e.g. Hagen et al., 1999), with the oldest record dating back to 1967. The Polish Academy of Sciences have measured mass balance stakes on Hansbreen (HBR) since 1989 (Grabiec et al., 2012). University of Oslo and the Norwegian Polar institute started
mass balance measurements on Etonbreen (ETN) on Austfonna in 2004 (e.g. Aas et al., 2016), while Uppsala and Utrecht universities initiated stake measurements on Nordenskiöldbreen (NSB) in 2006 (e.g. Van Pelt et al., 2012).

Observations of runoff from glaciated catchments are sparce, but daily simulated runoff is compared to available discharge measurements from two catchments in Svalbard: Bayelva and de Geerdalen.The total area of the Bayelva catchment is 31 km$^2$,

of which 54% is covered by glaciers. The discharge is measured using a pressure transducer and a float and wire system, which records the water level in a concrete-floored weir. The system is calibrated periodically to derive a rating curve that converts water level to discharge Killingtveit et al. (2003). The total area of the de Geerdalen catchment is 79 km$^2$, with 10% covered by glaciers. Discharge measurements are conducted in a narrow gorge with a stable bedrock profile using a similar system as for Bayelva Killingtveit et al. (2003). In early summer, discharge from both catchments is mainly from snowmelt, while in late summer, rainfall and glacier runoff contribute to the water flow. The monitoring at both stations is unattended, and thus the discharge data have periods with erroneous readings, mostly caused by ice or snow build-up at the sensor Killingtveit et al. (2003). However, the timing of discharge events is generally not affected.

### 3.4 Satellite observations

In addition to the in-situ measurements of mass balance performed by stake measurements, we use estimates of the geodetic mass balance for validation of the CryoGrid product. The geodetic mass balance is found by taking the difference between elevation data at different dates to find the change in volume. This volumetric change is then converted into mass balance by assuming a value for the bulk density. Unlike the climatic mass balance estimates provided in this study, geodetic mass balance includes frontal ablation from marine terminating glaciers. Therefore, we only compare our results to the geodetic balance of land-terminating glaciers.

Several studies have provided estimates of the geodetic mass balance of glaciers in Svalbard (e.g. Moholdt et al., 2010; Nuth et al., 2010; Morris et al., 2020), but here we use the estimate by Hugonnet et al. (2021) which used Advanced Spaceborne Thermal Emission and Reflection Radiometer (ASTER) imagery for determining the geodetic mass balance of all glaciers on Earth from 2000-2019. The results are available for all glaciers in the Randolf Glacier Inventory (Pfeffer et al., 2014) at a temporal resolution of 1, 2, 4, 5, 10, and 20 years. Here, we use the 5 year mass balance estimate for all land-terminating glaciers in Svalbard for model comparison (see Fig. 1).

## 4  CryoGrid community model

In this study we use and further develop the CryoGrid community model for simulations of the climatic mass balance and meltwater runoff. CryoGrid is an open-source model developed for climate-driven simulations of the terrestrial cryosphere. The model has a modular structure, with many different modules that can be added together in various combinations to represent a wide range of surface and subsurface conditions. Information about the different functionalities and structures are described in detail in Westermann et al. (2023).

Three modules are used to determine the stratigraphy of glaciers in Svalbard: a glacier (ice) module, a firn module and a snow module (see Fig. 2). The main components of each module are described below. All modules use the surface energy balance as an upper boundary condition. For simulations of seasonal snow, a simple ground module and a snow module is used (see Westermann et al. (2023) for details on the ground module).

**(a) Glacier setup**

**(b) Seasonal snow setup**

**Figure 2.** Evolution of CryoGrid stratigraphy for simulation of (a) glacier mass balance and (b) seasonal snow in this study. For glacier grid points (a), the model stratigraphy consists of up to three modules, which can be combined to represent the following four situations: glacial ice, glacial ice covered with snow, glacial ice covered by firn and snow, and glacial ice covered by firn. Between each module there is an interaction (IA) class, which determines the transfer of heat, water, and mass. A trigger function determines when modules can be added or removed from the stratigraphy. For seasonal snow (b), there is a ground module which can be coupled to a snow module.

## 4.1 Glacier module

The glacier module consists of layers of pure-ice with a user-defined constant ice thickness. This module has not been altered compared to the one described in Westermann et al. (2023). For this study, 47 layers with a thickness between 0.1 and 1 m were used, totaling 20 m of ice. Previous mass balance studies of Svalbard have used constant ice albedo values in the range of 0.3-0.4 (e.g. Østby et al., 2017; Van Pelt et al., 2019) for all of Svalbard. From calibration with available mass balance observations, we found the best results using an ice albedo of 0.4. When mass is removed from the model by runoff, evaporation or sublimation, mass is shifted up from an infinite ice reservoir below into the lowest model layer. This is done to prevent the glacier from disappearing during long spin-ups due to the lack of ice flow. The infinite reservoir is assumed to have the same temperature as the lowest model layer. If there is no snow on the surface, any excess water from rain or melt runs off instantaneously.

## 4.2 Snow and firn module

If snowfall is added to the model, a snow module is added on top of the glacier ice or firn (see Fig. 2). If the snow survives on the glacier surface for more than one year, the snow layer is moved to a firn scheme. The snow and firn scheme have the same model physics for this application, but newly fallen snow will not be mixed with a firn layer. These modules have been specifically added to the model for this study.The snow and firn modules follow a slightly altered CROCUS (Vionnet et al., 2012) snow scheme as described in Westermann et al. (2023). Some of the main differences to the snow schemes presented in Westermann et al. (2023) are:

- Additional output variables, including refreezing, internal accumulation, CMB, SMB.

- updated water percolation and runoff scheme, including a parameterisation for the hydraulic conductivity and a runoff timescale (described in Section 4.2.1)

- Regridding of layers below the surface (described in Section 4.2.2)

A brief description of some of the most important model physics (the albedo, temperature diffusion, and densification) which was not changed for this study is given in Supplement S1.

### 4.2.1 Water percolation and runoff

The water in a grid cell is either immobile and bound to the snow or firn, or it flows downwards driven by gravity. The limit between the two regimes is the irreducible water content $\theta_{fc}$, in this study chosen as 0.05. The vertical water flux $q_w$ [m s$^{-1}$] is therefore given by

$$
q_w = \begin{cases} -K & \text{for } \theta_w > \theta_{fc} \\ 0 & \text{for } \theta_w \leq \theta_{fc} \end{cases}
\tag{1}
$$

where $K$ is the hydraulic conductivity [m s$^{-1}$] and $\theta_w$ is the volumetric water content in the snow. In Westermann et al. (2023), a constants user-defined hydraulic conductivity is used. Here, the hydraulic conductivity is parameterized in terms of the snow grain diameter $d$ [m], the snow density $\rho_s$, and the effective liquid saturation $\Theta = (\theta_w - \theta_{fc})/(1 - \theta_{fc})$.

The hydraulic conductivity of snow is the product of the unsaturated conductivity, $K_r$, and saturated conductivity, $K_s$, i.e. $K = K_s K_r$. The saturated hydraulic conductivity (Shimizu, 1970) is given by

$$
K_s = 0.077 \frac{g}{v_w} d_g{}^2 exp(-0.0078\rho_s)
\tag{2}
$$

where $g$ is the gravitational acceleration [m s$^{-2}$] and $v_w = 1.787 \cdot 10^{-6}$ m$^2$ s$^{-1}$ is the kinematic viscosity of water. The unsaturated hydraulic conductivity (van Genuchten, 1980) is given by

$$
K_r = \Theta^{0.5} \left[ 1 - \left( 1 - \Theta^{1/(1-1/n)} \right)^{1-1/n} \right]^2
\tag{3}
$$

where the parameter $n$ is given by

$$n = 15.68 \exp(-460 d_g) + 1. \tag{4}$$

Water is not allowed to flow into an impermeable layer, here defined as layers with a density higher than $830 \, \mathrm{kg \, m^{-3}}$ (Cuffey and Paterson, 2010), or a layer that already has its entire pore space filled. Water which would have otherwise flowed into an impermeable layer becomes available to run off. For this study, we have added a delayed runoff scheme to Westermann et al. (2023). Runoff does not occur immediately, but depends on a characteristic local runoff scale $\tau_R$ [days] which increases with surface slope $S$ $[\mathrm{m \, m^{-1}}]$ as follows,

$$\tau_R = c_1 + c_2 \exp(-c_3 S) \tag{5}$$

where $c_1 = 0.33$ day, $c_2 = 25$ days, and $c_3 = 140$ (Lefebre et al., 2003). The runoff per timestep $R$ [m w.e.] is then calculated from the water in excess of the irreducible water content $W_{ex}$ [m], as

$$R = W_{ex} \frac{\Delta t}{\tau_R} \tag{6}$$

where $\Delta t$ is the time step in days. This delay in runoff means that water in excess of irreducible saturation may linger in a layer until it either refreezes or runs off. The irreducible water saturation is 0.05, following Vionnet et al. (2012), and the irreducible water content is thus 5% of the total porespace.

### 4.2.2 Vertical discretization

To avoid very thin snow layers, a simplified gridding scheme is used (Zweigel et al., 2021). During each timestep, new snow is added to the uppermost grid cell by calculating a weighted average between all variables describing the new and old snow (density, snow age, snow grain size etc). The water equivalent volume of snow is used as the weighting factor. When the top grid cell exceeds a target snow water equivalent (here 0.02 m) by more than 50%, it is split in two. If the top grid cell is smaller than 50% of the target snow water equivalent, it is merged with the below cell. The grid size of the top snow cell is therefore in the order of 0.01-0.03 m w.e. For deeper snow layers, the layer size doubles every 10 layers by splitting/merging of layers.

For the firn modules, the top layer has a maximum snow water equivalent thickness of 0.1 m, and the layer size doubles every 10 layers by merging/splitting of layers. Freshly fallen snow will always fall on top of the firn and never be mixed in with the top layer.

### 5 Results: Description and validation

This section first describes the significant trends in the meteorological variables produced by the CARRA data set, and then presents the validation of the forcing, glacial mass balance and runoff in both CryoGrid simulation against in-situ observations. Then, the results and trends in the the climate mass balance, runoff and refreezing is discussed for the CARRA-forced CryoGrid simulations. Finally, the AROME-ARCTIC simulations are evaluated and analysed against the CARRA-forced simulations.

## 5.1 Trends in CARRA meteorological variables

Figure 3 shows the average yearly temperature and precipitation in CARRA over 1991-2021, as well as significant trends (p<0.05) in both variables. The average temperature over Svalbard land areas is -7.9°C, with the highest average annual temperatures over low-elevation non-glaciated land (up to -2.0°C), and the lowest temperatures over high-elevation glacier points (down to -12.8°C). There is a significant positive trend in the temperature at all points ($p < 0.05$), with an average trend of $1.4°\mathrm{C\,decade}^{-1}$ ($p < 0.01$). The largest trends are in the east of Svalbard (up to $2.4°\mathrm{C\,decade}^{-1}$), while the lowest trend is along the west coast (down to $1.0°\mathrm{C\,decade}^{-1}$).

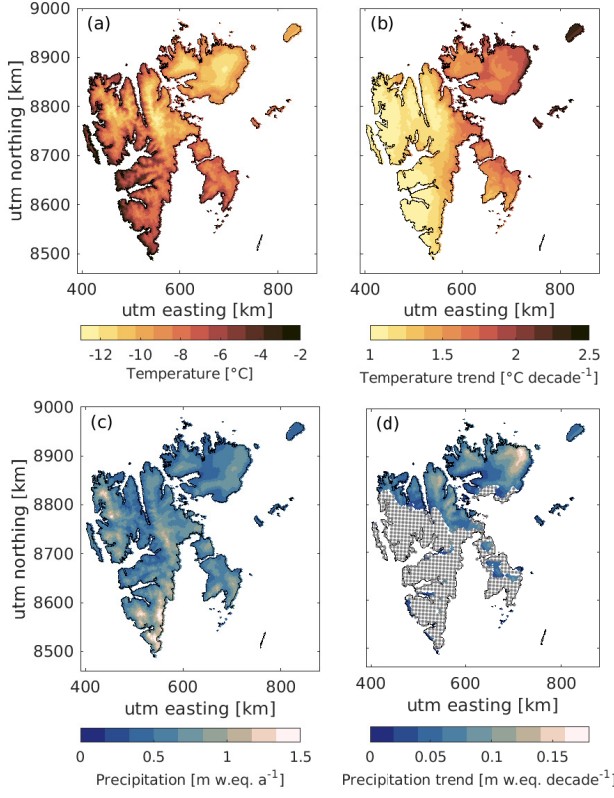

**Figure 3.** Average a) 2m-temperature and c) precipitation over 1991-2021 in CARRA. Significant ($p < 0.05$) b) temperature and d) precipitation trend in each points. Stippled areas have no significant trend.

The average precipitation over Svalbard is $0.62\ \mathrm{m\,w.e.\,yr}^{-1}$. There is a small but significant trend in the average yearly precipitation of $0.05\ \mathrm{m\,w.e.\,decade}^{-1}$ ($p < 0.01$). There is a larger trend in the precipitation over glacier-covered points ($0.06\mathrm{m\,w.e.\,decade}^{-1}$) than non-glacier-covered points ($0.03\mathrm{m\,w.e.\,decade}^{-1}$). Although there is no significant trend for all areas of Svalbard, there is a positive trend over e.g. Austfonna, Vestfonna, and North Spitsbergen. The largest trend is over NE Austfonna of $0.17\ \mathrm{m\,w.e.\,decade}^{-1}$. Over the investigated period, on average 90% of the precipitation fell as snowfall. There is a significant decreasing trend in the ratio between snow and rain, with the percentage of precipitation falling as snow

decreasing by -2.0% decade$^{-1}$ (p=0.01). For glacier-covered points, 95% of the precipitation falls as snow, with a significant decreasing trend of -1.3% decade$^{-1}$ (p=0.02). Over non glacier-covered points, however, 85% of the precipitation falls as snow, with a significant decreasing trend of -2.8% decade$^{-1}$ (p=0.01).

## 5.2 Evaluation

### 5.2.1 Forcing evaluation

The comparison of the CARRA forcing against observations from automatic weather stations shows a general good agreement. The MET Norway stations have been assimilated into the CARRA product, and it is therefore not surprising that there is a good agreement between the two. The largest differences in temperature are found for the Sveagruve II station ($\Delta T = -1.8°$C), but for most of the MET Norway stations the mean temperature difference is below $1°$C. The largest differences in relative humidity and wind speed are found at Kvitøya ($\Delta RH = 6.4\%$) and Pyramiden ($\Delta WS = -1.9 \, \mathrm{m \, s}^{-1}$), respectively.

The near-surface temperature at the glacier stations, which were not assimilated into the CARRA product, is generally well represented, with biases generally smaller than $1°$C. The exception is at the Etonbreen AWS, where CARRA has a cold bias. This can, however, partly be attributed to a warm bias in the AWS observations over time at this station due to sensor drift, before redundancy has been installed in 2016. The relative humidity has a maximum bias of 6.2%, while the wind speed bias ranges between -1.3 and $1.5 \mathrm{m \, s}^{-1}$. The incoming longwave and shortwave radiation in CARRA generally fits well with the observations, albeit with a small negative bias in the longwave radiation for most of the stations (ranging between -1.6 and -14W m$^{-2}$).

The evaluation of both forcing products against available AWS observations shows that the two products often provide similar results, but that the bias and root-mean-square-error of the CARRA product is generally smaller than for AROME-ARCTIC. For detailed evaluation of the model forcing against available AWS observations for both CARRA and AROME-ARCTIC, in addition to a discussion on the inter-comparison, we refer to Supplement S2,S3.

### 5.2.2 Mass balance evaluation

Mass balance, $b$, from stakes on eight glaciers in Svalbard is compared to the CryoGrid simulations of the surface mass balance in Table 2. Here, the surface mass balance is defined as the mass balance in the annual layer, and thus does not include refreezing in firn. The difference in mass balance, $\Delta b$, is defined as the modeled value minus the observed value at a given location.

Table 2 and Fig. 4 compare the stake observations and the nearest model gridpoint value for the CARRA-forced CRYOGRID simulations. Overall, there is a good agreement between the model and the observations, with biases and root-mean-square errors similar to (Van Pelt et al., 2019) or slightly better than (e.g. Østby et al., 2017) those found in other modeling studies. The largest difference in the winter mass balance occurs at Hansbreen (Fig. 4e) where the model has a large negative bias at all stake locations except at the lowest and highest elevations. There is also a negative bias in the summer mass balance at the low elevation stations, but there is good agreement at glacier stations at higher elevations. The largest average difference in summer occurs at Austre Brøggerbreen, where the CARRA-forced simulation underestimates the mass balance by 0.22 m w.e. yr$^{-1}$ on

**Table 2.** Evaluation of modelled results against observations from mass balance stakes in m w.e. yr$^{-1}$. The values from 1991/92-2017/18 shows the comparison with CARRA-forced model simulation, while for 2016/17-2017/18 the observations are compared to simulations using both CARRA and AROME-ARCTIC forcing. The results are given as CARRA forcing / AROME-ARCTIC forcing. Subscripts $w$, $s$, and $a$, respectively, refer to values calculated for winter months, summer months and annually.

| Period | Location | stakes | $\Delta b_w$ | rmse $b_w$ | $\Delta b_s$ | rmse $b_s$ | $\Delta b_a$ | rmse $b_a$ |
|---|---|---|---|---|---|---|---|---|
| | Austre Brøggerbreen | 3 | -0.18 | 0.21 | -0.22 | 0.47 | -0.41 | 0.60 |
| | Midtre Lovénbreen | 2 | -0.24 | 0.28 | 0.06 | 0.24 | -0.18 | 0.35 |
| | Kongsvegen | 9 | -0.08 | 0.17 | -0.11 | 0.28 | -0.19 | 0.35 |
| 1991/92 - | Hansbreen | 9 | -0.32 | 0.41 | -0.06 | 0.45 | -0.38 | 0.68 |
| 2017/18 | Holtedahlfonna | 10 | -0.04 | 0.13 | 0.05 | 0.25 | 0.007 | 0.30 |
| | Linnébreen | 1 | -0.15 | 0.17 | 0.18 | 0.25 | 0.03 | 0.19 |
| | Etonbreen | 7 | -0.05 | 0.12 | 0.04 | 0.19 | -0.02 | 0.21 |
| | Nordenskiöldbreen | 11 | 0.12 | 0.21 | -0.08 | 0.42 | 0.03 | 0.47 |
| | **Total** | - | **-0.08** | **0.21** | **-0.01** | **0.31** | **-0.09** | **0.39** |
| | Austre Brøggerbreen | 1 | -0.11 / -0.11 | 0.11 / 0.12 | -0.32 / -0.23 | 0.32 / 0.27 | -0.43 / -0.34 | 0.43 / 0.36 |
| | Midtre Lovénbreen | 2 | -0.19 / -0.21 | 0.20 / 0.24 | -0.09 / -0.10 | 0.16 / 0.22 | -0.28 / -0.31 | 0.29 / 0.32 |
| 2016/17 - | Kongsvegen | 6 | -0.02 / -0.08 | 0.09 / 0.13 | -0.29 / -0.27 | 0.34 / 0.33 | -0.32 / -0.35 | 0.43 / 0.37 |
| 2017/18 | Holtedahlfonna | 10 | 0.12 / 0.10 | 0.17 / 0.16 | -0.05 / -0.04 | 0.26 / 0.29 | 0.06 / 0.05 | 0.31 / 0.30 |
| | Etonbreen | 7 | -0.02 / 0.02 | 0.07 / 0.07 | 0.12 / 0.14 | 0.16 / 0.19 | 0.11 / 0.16 | 0.14 / 0.19 |
| | Nordenskiöldbreen | 11 | 0.24 / 0.32 | 0.29 / 0.37 | -0.19 / 0.15 | 0.46 / 0.46 | 0.05 / 0.47 | 0.48 / 0.65 |
| | **Total** | - | **0.06 / 0.07** | **0.18 / 0.22** | **-0.11 / -0.14** | **0.29 / 0.35** | **-0.05 / -0.10** | **0.33 / 0.42** |

average. Note, however, that both Hansbreen and Austre Brøggerbreen are small glaciers in complex topography, and thus may not be well represented by the the 2.5 x 2.5 km resolution of these simulations.

Table 2 also contains the comparison between the stake observations and the nearest model gridpoint for the AROME-ARCTIC-forced simulations. Only the 2016/17 and 2017/18 glaciological years were used for this evaluation. Overall, the CARRA and AROME-ARCTIC forced simulations perform almost equally well over these two years, with similar biases and root-mean-square errors for both the summer and winter balance. There is, however, a larger difference between the estimates of the annual mass balance, primarily due to large differences in the simulations for Nordenskiöldbreen when using the different forcings. For the CARRA forced runs for 2016/17-2017/18, the overestimation in the mass balance of Nordenskiöldbreen during the winter is balanced by excess melt during the summer, leading to only a small bias in the annual comparison. Using AROME-ARCTIC, the mass balance of Nordenskiöldbreen is underestimated both in summer and winter, leading to a large bias and rmse in the annual comparison. Nordenskiödbreen experiences a very strong accumulation-elevation gradient, due to high wind speeds and snow drift at lower elevations and calmer conditions at higher elevations. It is therefore difficult to accurately simulate this glacier without including snow re-distribution between grid points.

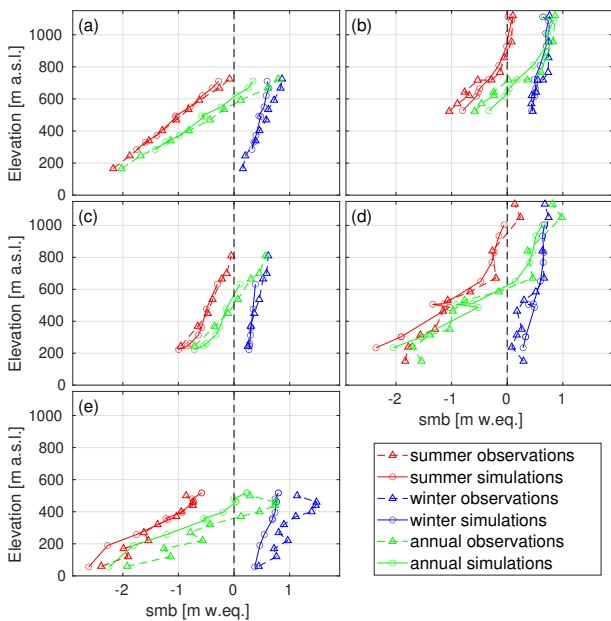

**Figure 4.** Average simulated (CARRA) and observed mass balance from 1991-2018 at each stake location for (a) Kongsvegen, (b) Holtedahl-fonna, (c) Etonbreen, (d) Nordenskiöldbreen, and (e) Hansbreen.

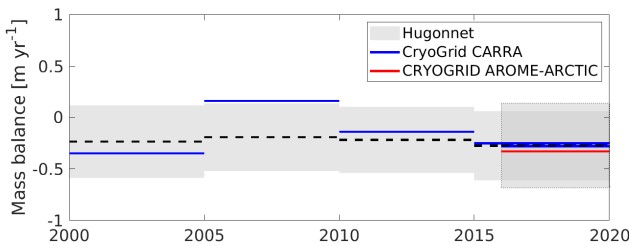

**Figure 5.** Geodetic mass balance of land terminating glaciers (from Hugonnet et al. (2021)) compared to the climatic mass balance simulated in CryoGrid.

In addition to the in-situ mass balance, we use estimates of the geodetic mass balance of land-terminating glaciers by
370 Hugonnet et al. (2021) to validate the mass balance results. Since the geodetic estimates include refreezing below the annual layer, we here use the climatic mass balance for the comparison. Figure 5 compares the CMB from CryoGrid for five year periods between 2000 and 2020 against estimates from Hugonnet et al. (2021). The simulated CMB is within the uncertainty estimate of the geodetic data for the whole period, except in 2005-2009 where the CMB is slightly higher than the uncertainty estimate (by $0.02 \, \text{m w.e. yr}^{-1}$). The AROME-ARCTIC-forced simulations are within the uncertainties of the geodetic estimate,
but have a slightly lower mass balance than the CARRA-forced simulations for the same period (-0.29 $\text{m w.e. yr}^{-1}$ using CARRA vs. -0.34 $\text{m w.e. yr}^{-1}$ using AROME-ARCTIC.)

### 5.2.3 Runoff evaluation

Comparison of the yearly observed and modelled discharge were conducted for the CARRA-forced simulations for Bayelva and de Geerdalen catchments. To evaluate the accuracy of the model simulations, we calculate the Nash-Sutcliffe efficiency (NSE), percent bias, and ratio of the root mean square error to the standard deviation of measured data (RSR). Moriasi et al. (2017) Moriasi et al. (2007) suggested that discharge models can be deemed sufficient if the NSE > 0.5, the percent bias is within ±25% and the RSR < 0.7. For Bayelva, we find that there is a good agreement between the simulations and observations based on all parameters. There is a positive percentage bias of 9.0%, while the NSE is 0.71 and the RSR is 0.54. We also find a good agreement for de Geerdalen, with a positive percentage bias of 6.6%, NSE of 0.65 and RSR of 0.60.

Although no routing model is used for these simulations to take into account the time delay in discharge, there is still a high correlation $r$ in the daily runoff of 0.88 and 0.86 for Bayalva and de Geerdalen, respectively.

### 5.3 CARRA-forced simulations

### 5.3.1 Climatic mass balance

The area-averaged climatic mass balance of all Svalbard glaciers for the whole CARRA simulation period is found to be -0.08 m w.e. $yr^{-1}$ Fig. 6a-c show the annual, winter, and summer climatic mass balance over Svalbard. The results are shown for each mass balance year, here defined as September to August. For calculations of the winter and summer mass balance, we use fixed dates of 1 April and 1 September. The most negative values are found at low-elevation areas in S and SW Spitsbergen (Fig. 6a), with the CMB reaching down to -3.23 m w.e. $yr^{-1}$, while the most positive values are found at high-elevation areas in central Spitsbergen, reaching a maximum CMB of 1.16 m w.e. $yr^{-1}$. The winter mass balance (Fig. 6b) is on average positive at all points, while the summer mass balance (Fig. 6c) is negative except at some high-elevation points in NE and NW Spitsbergen.

Figure 6d shows the temporal evolution of the summer, winter, and annual CMB. The most positive CMB (0.43 m w.e. $yr^{-1}$) was found in the 2007/08 mass balance year, while the most negative CMB (-0.68 m w.e. $yr^{-1}$) was found in 2019/20.

The winter CMB is on average 0.44 m w.e. $yr^{-1}$, with a maximum in 2015/16 (0.65 m w.e. $yr^{-1}$) and a minimum in 2001/02 (0.28 m w.e. $yr^{-1}$). The summer CMB is on average -0.52 m w.e. $yr^{-1}$, with the most negative value in 2020 (-1.0 m w.e. $yr^{-1}$) and the least negative value in 2008 (-0.19 m w.e. $yr^{-1}$). There is no significant trend in winter, summer, or annual CMB over the investigated period.

Figure 7 shows the winter (blue bars), summer (red bars), and annual CMB (green line) for eight different regions of Svalbard. The glaciers in Nordenskiöldland have the most negative annual CMB (-0.73 m w.e. $yr^{-1}$), with glaciers losing mass during all years except 2007/08. The most positive average CMB is in NE Spitsbergen (0.11 m w.e. $yr^{-1}$). For all areas, 2012/13 was a year with a strongly negative summer CMB. In 2019/20, NW-Spitsbergen experienced a record amount of melt (-1.37 m w.e. $yr^{-1}$) in combination with a record low winter CMB (0.18 m w.e. $yr^{-1}$). Most other regions also experienced strong summer melt, with the exception of Edgeøya and Barentsøya where the summer CMB is close to the average over the simulation period.

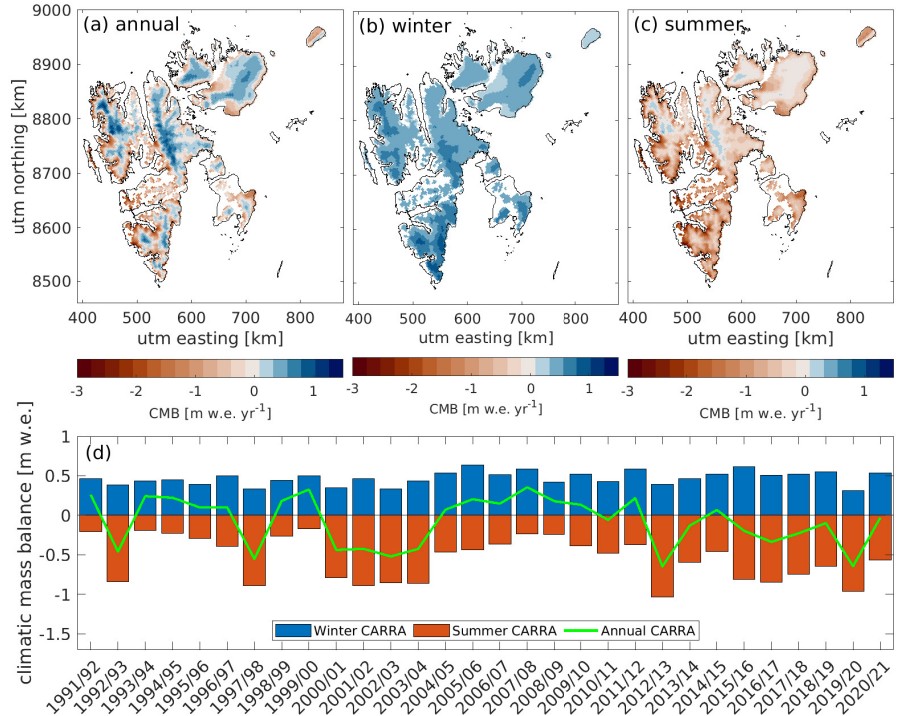

**Figure 6.** Climatic mass balance of Svalbard from 1991/92 to 2020/21. The top row contains maps of the average (a) annual, (b) winter, and (c) summer CMB, while (d) shows the temporal evolution of the summer, winter and annual CMB for each mass balance year (September - August).

Kvitøya, Barentsøya/Edgeøya, and Nordenskiöldland all have a significant negative trend in the annual CMB of -0.17, -0.22, and -0.27 $\mathrm{m\,w.e.\,decade^{-1}}$, respectively. The other regions also have negative trends, but they are not significant at a 95% confidence interval. There is a small, but significant, positive trend in the winter CMB of 0.05 $\mathrm{m\,w.e.\,decade^{-1}}$ for both Austfonna and Vestfonna, but no significant trend is found for the other areas. Kvitøya (-0.19 $\mathrm{m\,w.e.\,decade^{-1}}$), S Spitsbergen (-0.18 $\mathrm{m\,w.e.\,decade^{-1}}$), and Nordenskiöldland (-0.22 $\mathrm{m\,w.e.\,decade^{-1}}$) have significant negative trends in the summer balance.

### 5.3.2 Refreezing

Refreezing is defined as all liquid water that refreezes within snow and firn, without taking into account that this may melt again. The average annual refreezing for glacier-covered and land areas is 0.24 $\mathrm{m\,w.e.\,yr^{-1}}$ and 0.11 $\mathrm{m\,w.e.\,yr^{-1}}$, respectively. The lowest annual refreezing is simulated at low elevations, where only thin seasonal snowpacks are present, thus limiting the amount of refreezing. The largest refreezing is found in the accumulation zones (Fig. 8), where average values up to 0.32 $\mathrm{m\,w.e.\,yr^{-1}}$ are simulated. In these areas, water can percolate down into firn layers and refreeze over the winter season. The spatial distribution of this internal accumulation (defined as refreezing beneath the annual layer) is shown in Fig. 8b,

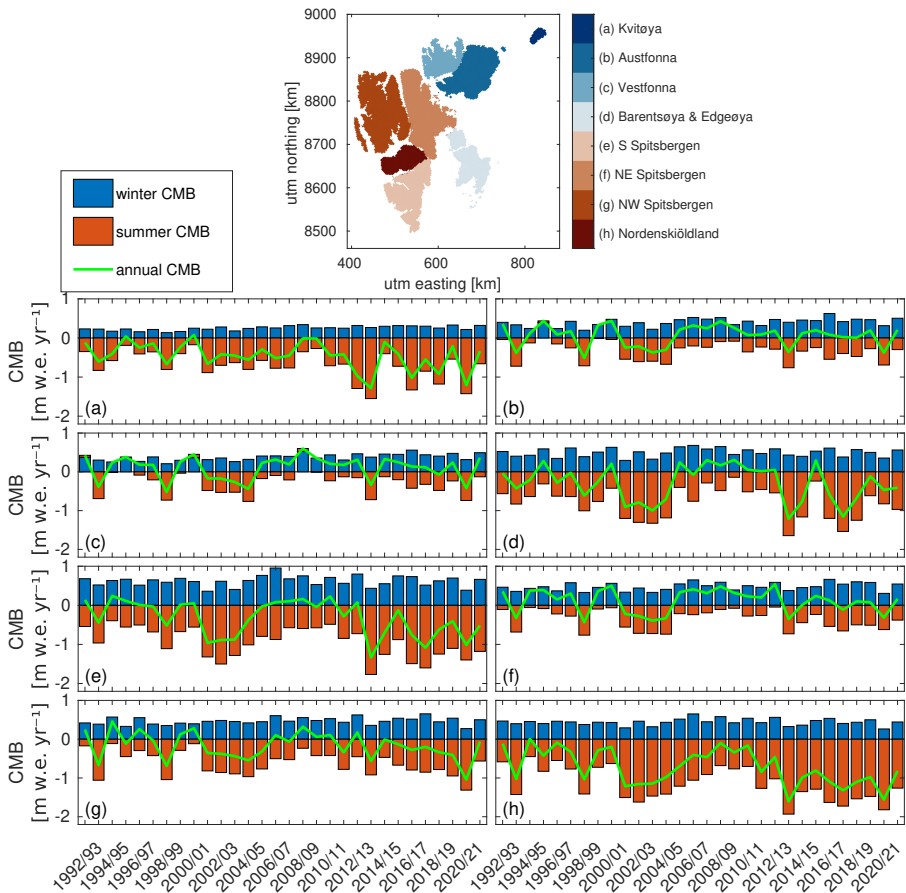

**Figure 7.** Climatic mass balance for different regions of Svalbard. Dark red bars show the summer balance and dark blue bars show the winter, while the green line is the yearly CMB.

showing that a significant fraction of the refreezing at higher elevations occurs in deeper layers. The average annual internal accumulation is $0.11$ m w.e. $yr^{-1}$, and thus accounts for almost half of the total refreezing (Fig. 8c). There is a significant

negative trend in the refreezing within glaciers of -13 mm w.e. $decade^{-1}$ (p < 0.01), which is primarily due to a decrease in internal accumulation.

For glacier-covered areas, annual refreezing of melt and rainwater accounts for $25\%$ of the total accumulation, varying between $19\%$ and $32\%$ over the simulation period. There is a significant negative trend in the contribution of refreezing to the total accumulation of -2.1% $decade^{-1}$ (p < 0.01).

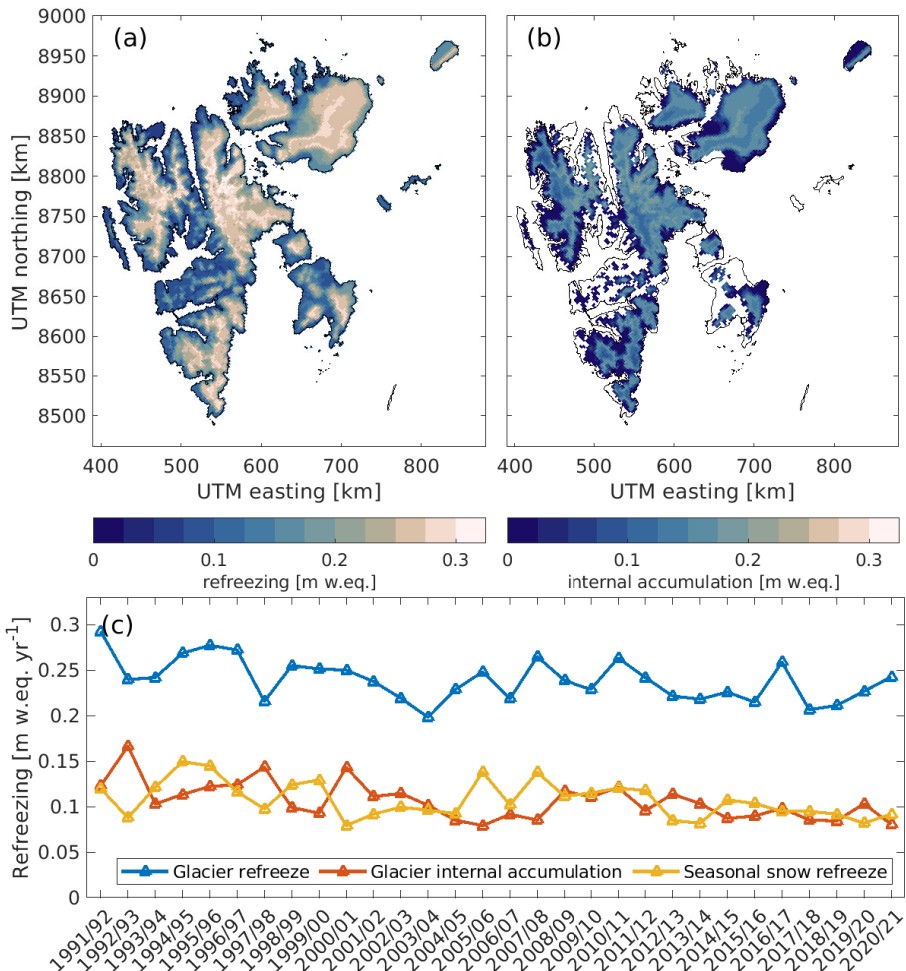

**Figure 8.** (a) Average annual refreezing for glaciers and seasonal snow on non-glaciated land areas. (b) Average annual internal accumulation from glacier-covered areas. (c) The temporal variation in refreezing and internal accumulation for glaciers and seasonal snow.

### 5.3.3 Runoff

The simulated runoff from both glacier-covered and non-glaciated land points are shown in Fig. 9. The average runoff for glaciers has a similar pattern as the CMB (Fig. 6), with the highest runoff in low-elevation regions (up to $3.0 \mathrm{~m~w.e.~yr}^{-1}$) and lowest runoff in high-elevation areas in central Spitsbergen (down to $0.02 \mathrm{~m~w.e.~yr}^{-1}$).

The average total runoff from glacier-covered regions is $0.80 \mathrm{~m~w.e.~yr}^{-1}$ ($29 \mathrm{~Gt~yr}^{-1}$) and for land regions it is $0.50$

$\mathrm{m~w.e.~yr}^{-1}$ ($12 \mathrm{~Gt~yr}^{-1}$). While there is a large variation in runoff from glacier-covered regions, the runoff from land areas is relatively stable throughout the whole period (Fig. 9c). The minimum and maximum runoff from seasonal snow occurred in 1996 ($0.36 \mathrm{~m~w.e.~yr}^{-1}$) and 2016 ($0.64 \mathrm{~m~w.e.~yr}^{-1}$), respectively.

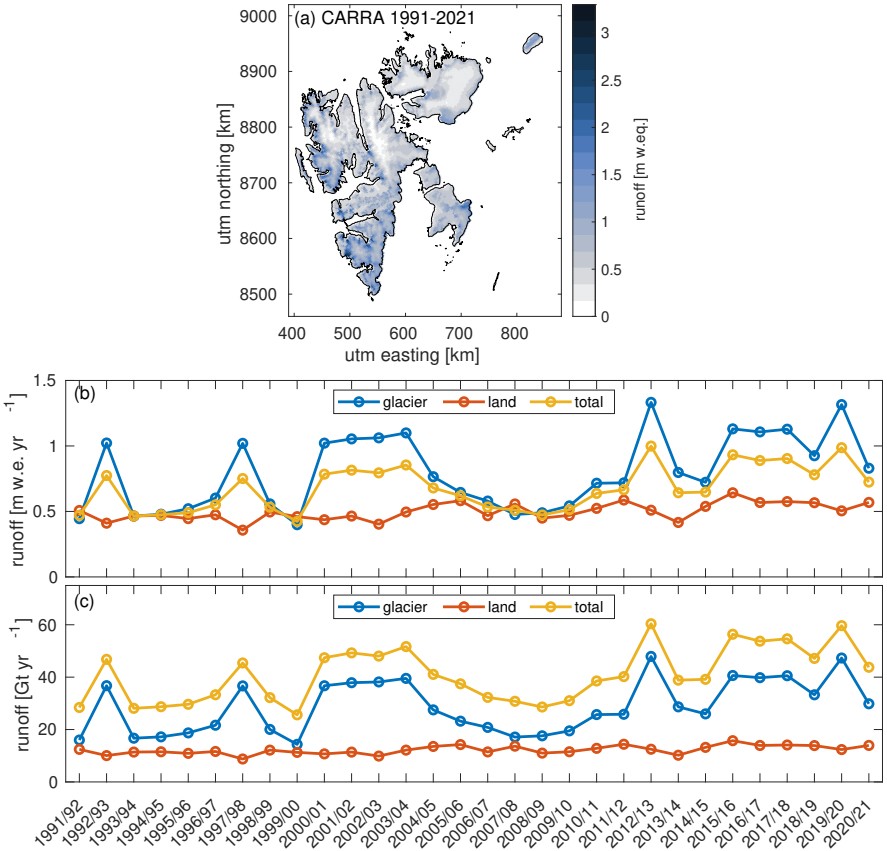

**Figure 9.** a) Average runoff over the whole CARRA simulation period (1991/92–2020/21). b+c) Timeseries of runoff from glaciers and seasonal snow on non-glaciated land in b) m w.e. yr$^{-1}$ and c) Gt yr$^{-1}$.

For glacier covered areas, the minimum runoff occurred in 2008 (0.40 m w.e. yr$^{-1}$), where the discharge was almost equal to that coming from seasonal snow. The runoff during this year was low for the entire Svalbard area, with particularly low rates along the Western coast. The largest runoff from glaciers occurred in 2013 (1.33 m w.e. yr$^{-1}$), closely followed by 2020 (1.31 m w.e. yr$^{-1}$). In 2013 there was generally high runoff over the entire peninsula compared to the average values, with especially large runoff rates in southern Spitsbergen and Barentsøya.

There is a significant, positive trend in both the glacier runoff and the runoff from seasonal snow of 0.14 m w.e. decade$^{-1}$ (5.2 Gt decade$^{-1}$, p=0.01) and 0.04 m w.e. yr$^{-1}$ (1.1 Gt decade$^{-1}$, p<0.01), respectively. The runoff from land is, of course, determined by the amount of precipitation in the forcing product. An increase in the runoff from seasonal snow shows that the precipitation over non-glacier covered areas is increasing.

## 5.4 AROME-ARCTIC forced simulations

In this section, it is investigated if AROME-ARCTIC simulations can be used to extend the CARRA-forced simulations and provide almost real-time estimates of the conditions of glaciers on Svalbard.

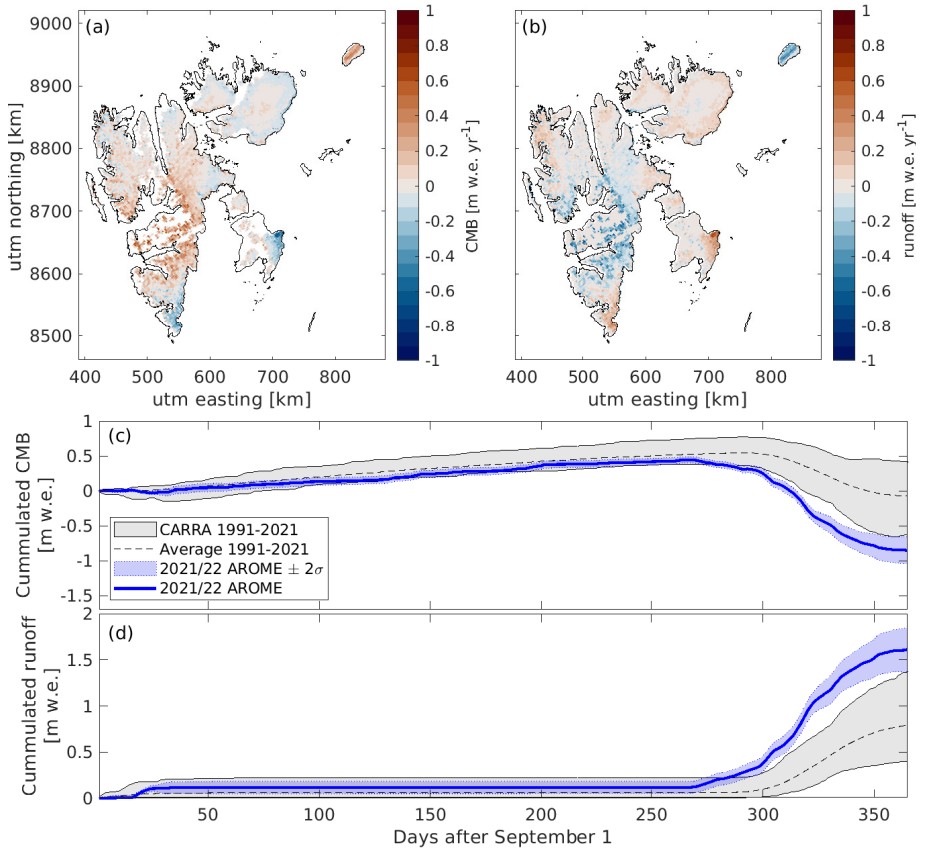

**Figure 10.** The average difference in simulated (a) CMB and (b) runoff for 2016-2021 when using CARRA and AROME-ARCTIC forcing. The values are given as AROME-ARCTIC-forced results minus CARRA-forced results. (c+d) The span in daily accumulated (c) CMB and (b) runoff from 1991/92-2021/22 simulated using CARRA climate forcing. The 2021/22 mass balance years are simulated using AROME-ARCTIC (shown in red). The uncertainty of the AROME-ARCTIC estimate is defined as two standard deviations of the differences between the CARRA and AROME-ARCTIC forced simulations from 2016/17-2020/21.

Figure 10a-b shows the average difference between the CARRA-forced and AROME-ARCTIC-forced CMB and runoff over the 2016-2021 period. For most regions, the CMB (Fig. 10a) simulated using AROME-ARCTIC closely matches that of CARRA, although with large deviations for Nordenskiöldland. For the other regions, the average difference between the CARRA and AROME-ARCTIC estimates is $< 0.10 \ \mathrm{m \, w.e. \, yr^{-1}}$, while for Nordenskiöldland it is $0.41 \ \mathrm{m \, w.e. \, yr^{-1}}$.

Averaged over the whole domain, the annual CMB is similar in the two simulations, but with a more negative CMB when
using AROME-ARCTIC of about -0.1 $\mathrm{m \, w.e. \, yr^{-1}}$ from 2016-2017, and a more positive CMB when using AROME-ARCTIC

in 2019-2021 of approximately 0.1 m w.e. yr$^{-1}$. Generally, the AROME-ARCTIC simulations contain slightly lower winter CMB, with an average difference of -0.04 m w.e. yr$^{-1}$. The summer CMB is more variable, but generally the values in the AROME-ARCTIC simulations are less negative than CARRA (by -0.05 m w.e. yr$^{-1}$ on average).

The glacier runoff (Figure 10b) is generally higher in the AROME-ARCTIC-forced simulations for SW Spitsbergen and Barentsøya/Edgeøya, and lower for Kvitøya and Nordenskiöldland. The average difference (AROME-ARCTIC - CARRA) from 2016-2021 is -0.03 m w.e. yr$^{-1}$, ranging between 0.10 and -0.13m w.e. yr$^{-1}$ for individual years. The runoff from seasonal snow on non-glaciated land is also similar overall, with the average value from AROME-ARCTIC only slightly larger than CARRA by 0.008m w.e. yr$^{-1}$. Interestingly, although the glacier runoff is lower in the AROME-ARCTIC simulations for Nordenskiöldland, the runoff from seasonal snow is not. This indicates that the difference between the CARRA and AROME-ARCTIC runoff estimates is not due to differences in precipitation.

Thus, AROME-ARCTIC forcing generally produces results for the mass balance and runoff of Svalbard that is similar, within 0.2 m w.e. yr$^{-1}$ for both variables, to that simulated using CARRA forcing. This indicates that AROME-ARCTIC can be used to create near real-time estimates of the climatic mass balance of Svalbard, although the uncertainties may be larger than generated by the CARRA-forced simulations. However, for simulations of Nordenskiöldland, one should be aware that large differences exist compared to CARRA.

Figure 10c-d shows the cumulative CMB and glacier runoff for the 2021/22 glaciological year simulated with AROME-ARCTIC forcing compared to the span in simulations from the 1991-2021 CARRA-forced simulations. The mean of the CARRA-forced simulations from 1991-2021 is shown with a dashed black line, while the minimum and maximum years are shown in grey. In order to better compare the CARRA and AROME-ARCTIC forced simulations, the AROME-ARCTIC-forced estimates are shown with an uncertainty, given as two standard deviations of the differences between the CARRA and AROME-ARCTIC forced simulations from 2016/17-2020/21. In other words, this uncertainty indicates what the CMB would most likely be if CARRA had been used as forcing as opposed to AROME-ARCTIC for these simulations.

During the winter months, there is generally little difference between the simulations with the difference forcings, and we can therefore have a high confidence in the AROME-ARCTIC results. During the summer, however, larger differences arise between the different products, which accumulate over the melt season. Based on the 2016/17-2020/21 simulations, the estimated standard error in both the CMB and runoff is 0.12 m w.e. yr$^{-1}$ by the end of the mass balance year.

Based on AROME-ARCTIC, 2021/22 is a record negative mass balance year for Svalbard, with a CMB of -0.86 m w.e. yr$^{-1}$. There is a highly negative mass balance in all regions in Svalbard. The runoff from glaciers is also the highest over the simulation period, of 1.6 m w.e. yr$^{-1}$ (58 Gt yr$^{-1}$).

## 6 Discussion

### 6.1 Frontal ablation

The datasets presented in this paper only account for the climatic mass balance, and therefore do not include e.g. the mass loss from frontal ablation. However, by comparing the climatic mass balance estimate to the estimates of total mass balance

from e.g. geodetic methods, one can reach a rough estimate of the mass loss due to calving. Similar to the model validation, we use the estimates from Hugonnet et al. (2021) but now include tidewater glaciers in the comparison. By subtracting the CARRA-forced simulated climatic mass balance from the geodetic estimate, we can get an estimate of the calving rate. These estimated calving rates are from 0 to -0.19 $\mathrm{m\,yr^{-1}}$ in the early 2000s (2000-2004), followed by an increase in 2005-2009 with possible values between -0.15 and -0.67 $\mathrm{m\,yr^{-1}}$. These numbers are consistent with the estimate by Błaszczyk et al. (2009) of -0.18 $\mathrm{m\,yr^{-1}}$ from 2000-2006. In the first half of the 2010s, the calving rate is between 0 and -0.43 $\mathrm{m\,yr^{-1}}$, while in the latter half it is between 0 and -0.5 $\mathrm{m\,yr^{-1}}$. The large range in calving rates reflects the uncertainty in the geodetic estimate. The calving after 2010 is likely increased due to the surge of Basin-3, the largest outlet basin of the Austfonna ice cap, which significantly increased the calving from the ice cap (e.g. Dunse et al., 2015).

## 6.2 Uncertainty

Several sources of uncertainty are introduced through the creation of the glacier mass balance dataset in this study. The sources of these uncertainties are comprised of the model physics, the initial model state, atmospheric forcing, glacier extent and topographic simplification. It is, however, difficult to quantify the contribution of each individual source. This section discusses these sources of uncertainty.

### 6.2.1 Model physics and initialisation

Although the snow and firn scheme is based on the CROCUS model, the physics of which has been used and validated in a number of glacier mass balance and snow studies (Cullather et al., 2016; Schmidt et al., 2017; Verjans et al., 2019), there may still be uncertainties connected to using this model in Svalbard. For example, previous studies have shown that CROCUS does not always perform well under Arctic conditions, therefore, we have made a number of changes to the original model as e.g. suggested by Royer et al. (2021). However, most of the model parameters used by the snow and firn scheme are based on recommendations from previous studies and have not been tuned for the conditions of Svalbard. Although the model does well when compared to observations, potential biases may arise in other regions of Svalbard.

In addition, we use a constant ice albedo in the model, which could be a major simplification given that the ice albedo varies across Svalbard from 0.15 to 0.44 (Greuell et al., 2007). In future work, this could be improved by using estimates of the ice albedo from e.g. MODIS observations to create a map of the ice albedo (Schmidt et al., 2017) and/or updating the albedo parameterisation to account for dust and impurity content.

To initialize the sub-surface conditions, a 30-year spin-up was performed. This was done by repeating the forcing from 1991-2000, until the model output was approximately in balance with the applied climate forcing. This could introduce some biases in both the extent and depth of the firn area, as the glaciers may not have been in balance with the 1991-2000 climate in reality.

### 6.2.2 Model forcing

We find good agreement between CARRA forcing and the meteorological variables and the incoming radiation over glaciers and non-glaciated land (see Supplement S2,S3 for details), although it should be noted that the non-glaciated land based AWSs are assimilated into the CARRA product. Comparison against winter mass balance stake observations shows CARRA precipitation has a low rmse overall ($0.21$ m w.e. $\mathrm{yr}^{-1}$), but is slightly underestimated over most of the glaciers (Table 2). This could be partly due to the spatial resolution of CARRA, as at a 2.5 km resolution the model might miss some of the impact of the

terrain on the precipitation distribution, particularly in areas with complex topography. In addition, the simulated mass balance representing a 2.5 x 2.5 km cell may not be directly comparable with point-observations, as heterogenities in the energy and mass balance occur at spatial scales less than 2.5 km. For example, in areas with high wind, redistribution of snow by the wind may have a large effect on the winter mass balance (e.g. Winther et al., 2003). Furthermore, since the stake observations are mainly taken along the glacier centerline, the observations do not reflect the horizontal distribution of the mass balance along

the measured glaciers.

In addition, using AROME-ARCTIC to generate a real time dataset adds additional uncertainties as this is a forecast and not a reanalysis product. From 2016 until the summer of 2019, the model was initialized with too little snow over some glacier points in the ablation area, thus leading to unrealistically high surface and 2m temperatures. To try to counter this effect, we use

the 10 m temperature for the AROME-ARCTIC-forced simulations when unrealistically high surface temperatures occur, but some biases may still persist. In addition, as previously discussed, due to missing data it is not always possible to use AROME-ARCTIC forecasts with a 6-hour lead time. Using an earlier forecast with longer lead times introduces higher forecast errors, and therefore using forecasts at different time steps may give different results.

As an example, Fig. 11 shows the differences in incoming radiation between various forecast lead times during August 2019.

Fig. 11a shows the mean difference between 6 hour and 24 hour lead times. Overall, the mean absolute difference is small (1.5 $\mathrm{W\,m}^{-2}$) with a maximum average deviation of 7.0 $\mathrm{W\,m}^{-2}$. However, at any given location and time step, the difference in incoming radiation between the different lead times may be a large as 335 $\mathrm{W\,m}^{-2}$. An example of the temporal difference between forecast lead times of 6, 12, 18 and 24 hours is shown in the Fig. 11b. The grey area shows the maximum and minimum differences between the incoming radiation with a 6 hour lead time and 12, 18, and 24 hour lead times. The mean differences

between the 6-hour and 12/18/24-hour lead times over the whole month is small, ranging between -8.2 to 8.4 $\mathrm{W\,m}^{-2}$, but at any given time step large differences ($> 100$ $\mathrm{W\,m}^{-2}$) occur. Similar effects can be seen in the other meteorological variables, like precipitation, wind speed, and temperature. We expect the effect of the lead time used to be small over monthly or yearly timescales, but it can introduce large errors for specific days or areas.

Furthermore, the re-gridding of the AROME-ARCTIC product to the CARRA grid using linear interpolation may introduce

additional errors.

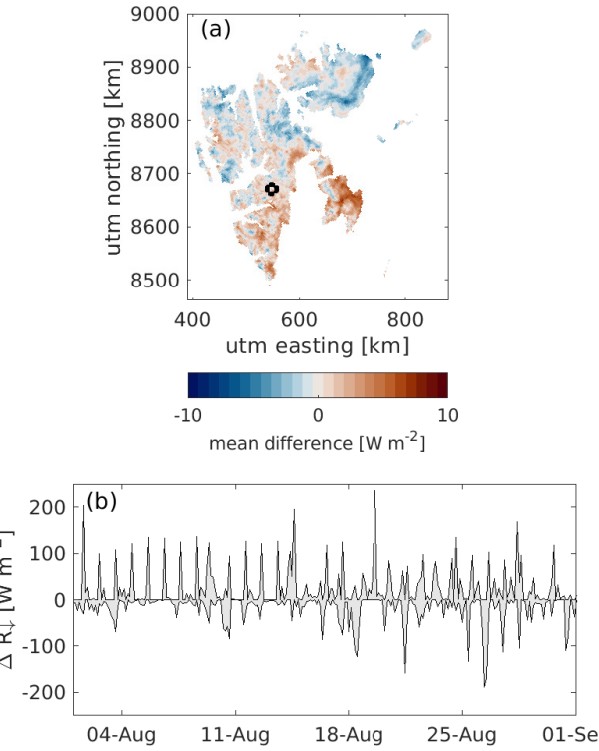

**Figure 11.** (a) Mean difference in August incoming radiation (R↓) between 6-hour and 24-hour forecast lead times. (b) The temporal difference in incoming radiation between 12, 18, and 24 hour lead times versus a 6 hour lead time (grey area). Location of point is shown with a circle in (a).

### 6.2.3 Glacier extent and topography

Throughout the simulation period, we assume the elevation and glacier mask is fixed thus neglecting the effect of ice flow and elevation changes on the mass balance. Both the elevation and the glacier mask are based on observations collected between 2000-2010 and should therefore be representative for most of the investigated period.

Using a fixed elevation mask may introduce a negative bias in the beginning of our study period, as the elevation may be too low, and a positive bias towards the end of the study period where the used elevation mask may be too high. On average for Svalbard, the glaciers elevation decreased at a rate of $0.36 \, \mathrm{m \, yr^{-1}}$ from 2000-2020 (Hugonnet et al., 2021), while between the mid-1960's and 2005 the glacier elevation outside Austfonna and Kvitøya decreased, on average, at a rate of $0.49 \, \mathrm{m \, yr^{-1}}$ (Nuth et al., 2010). Considering the elevation map used in this study is based on observations from the 2000s, we expect the

maximum average deviation to be 10 m. Assuming a change of mass balance with elevation of $3 \cdot 10^{-3} \, \mathrm{m \, w.e. \, m^{-1}}$, we expect the error associated with the constant glacier mask to be less than $0.03 \, \mathrm{m \, w.e. \, yr^{-1}}$.

The added error of a fixed glacier mask has previously been investigated by Østby et al. (2017). The authors found that the error in the climatic mass balance associated with using a fixed glacier mask (based on observations from the 2000s) as

opposed to a time-varying mask was on average $0.02 \, \mathrm{m\,w.e.\,yr^{-1}}$ for the period 1957-2014. Since the period investigated in
this study is smaller and more closely matches the time period the glacier mask was created, we expect the error due to a fixed
glacier mask in our simulations to be equal or smaller than the value found by Østby et al. (2017).

In addition, van Pelt et al. (2021) investigated the effect of ignoring both elevation and glacier mask changes on future
projections for Svalbard from 2018-2060. Over this time period, the authors found that the increased melt due to a lowering of
the glacier surface was nearly balanced by the melt reduction due to a changing glacier mask, and thus the introduced error in
the runoff and CMB was small.

## 6.3 Comparison with other studies

Several other studies have previously quantified the Svalbard-wide mass balance and runoff. Direct comparison between our
results and other studies is in some cases hampered by differences in time period, areal coverage, and the type of mass balance
calculated (e.g. estimates from gravimetry or geodetic methods will estimate the total mass balance, including frontal ablation).
Here, we only compare against studies which calculate either the climatic or surface mass balance, and who have published
results within our simulation period of 1991-2020. When available, we compare simulated average 2m temperature, yearly
precipitation, climatic mass balance, and runoff (Table 3 and Fig. 12).

| study | period | Tair [°C] | precipitation [m w.e. yr⁻¹] | CMB [m w.e. yr⁻¹] | runoff [m w.e. yr⁻¹] |
|---|---|---|---|---|---|
| Lang et al. (2015) | 1998-2007 | - | 0.56 | -0.088 | - |
| This study | 1998-2007 | - | 0.64 | -0.073 | - |
| Aas et al. (2016) | 2003-2013 | - | - | -0.26 | - |
| This study | 2003-2013 | - | - | -0.038 | - |
| Østby et al. (2017) | 1991-2014 | -7.3 | 0.70 | -0.10 | - |
| This study | 1991-2014 | -8.4 | 0.65 | -0.057 | - |
| Van Pelt et al. (2019) | 1991-2018 | -8.5 | 0.95 | 0.015 | 0.80 |
| This study | 1991-2018 | -8.0 | 0.62 | -0.077 | 0.79 |
| Noël et al. (2020) | 1991-2018 | - | 0.71[1] | -0.064 | 0.77 |
| This study | 1991-2018 | - | 0.70[1] | -0.077 | 0.79 |

**Table 3.** Svalbard climatic variables (precipitation and 2m temperature), climatic mass balance, and glacier runoff from different modeling
studies. [1]only precipitation over glaciers.

The climatic mass balance simulated in this study is similar to estimates from other studies. The CMB in this study is
slightly less negative than that simulated by Lang et al. (2015), which could partly be due to the higher precipitation in this
study. There are larger differences between our CMB and the estimates by Aas et al. (2016) and Østby et al. (2017), where our
simulated CMB is higher by 0.22 and $0.05 \, \mathrm{m\,w.e.\,yr^{-1}}$, respectively. In the case of Østby et al. (2017), this is partly due to a

big difference between the estimates for 2013, where the CMB estimated by Østby et al. (2017) is strongly negative. A more negative mass balance is consistent with the higher average 2m temperatures used for their simulations. On the other hand, our simulations have a more negative CMB than those simulated by Van Pelt et al. (2019). This is likely related to the much lower

precipitation in our simulations. When comparing the estimates of specific runoff, our results are consistent with Van Pelt et al. (2019). The precipitation, CMB, and runoff values in this study are consistent with those of Noël et al. (2020).

The temporal evolution of each of the CMB studies are plotted against our results in Fig. 12. The temporal pattern is similar for all the estimates, with high inter-model correlations between 0.8 and 0.9.

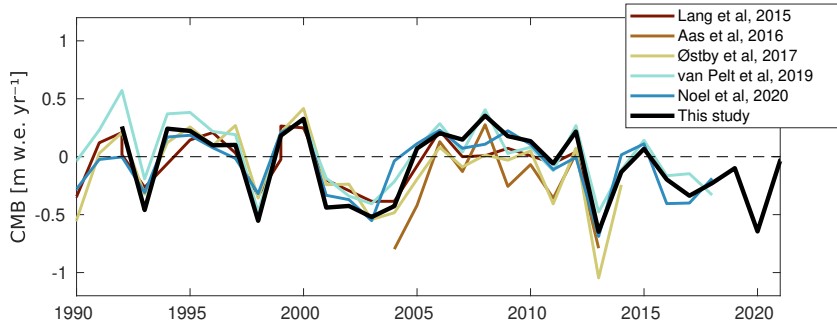

**Figure 12.** Time series of yearly CMB from different model studies from 1990-2021.

## 7   Conclusions

Using the novel high-resolution reanalysis dataset CARRA as well as the high-resolution regional forecast product AROME-ARCTIC as forcing for simulations of the coupled energy balance-subsurface model CryoGrid, we performed high-resolution simulations of the mass balance and runoff for Svalbard. The results from both the CARRA and AROME-ARCTIC forced simulations are presented, and the results are validated against in-situ observations from automatic weather stations and mass balance stakes as well as geodetic estimates.

We find that the area-averaged climatic mass balance over the period is slightly negative at -0.08 $\mathrm{m\,w.e.\,yr^{-1}}$. There is no statistically significant trend in the climatic mass balance over the investigates period. The average glacier runoff from 1991/92-2020/21 is 0.80 $\mathrm{m\,w.e.\,yr^{-1}}$, while the runoff from non-glaciated land is 0.50 $\mathrm{m\,w.e.\,yr^{-1}}$. There is a significant positive trend in both the glacier runoff (0.14 $\mathrm{m\,w.e.\,decade^{-1}}$) and land runoff (0.04 $\mathrm{m\,w.e.\,decade^{-1}}$). The timing and amount of freshwater runoff from Svalbard has important implications for the ecosystems in the surrounding fjords. Changes

is freshwater discharge affect a wide range of physical, chemical, and biological processes, including e.g. fjord circulation (Carroll et al., 2017), light availability (Hop et al., 2002; ARIMITSU et al., 2012), water biogeochemisty (Wadham et al., 2013; Bhatia et al., 2013), and marine primary production (Juul-Pedersen et al., 2015; Hopwood et al., 2020). Freshwater from tidewater glaciers may affect these processes in a different manner from seasonal snow runoff or runoff from land-terminating glaciers, and it is therefore important to quantify the amount of different types of runoff.

For the 2016/17 - 2010/21 glaciological years, the area-averaged CMB in the AROME-ARCTIC-forced simulations differed by up to $0.1\ \mathrm{m\,w.e.\,yr^{-1}}$ from the CARRA-forced simulations. The largest differences were found in Nordenskiöldland, with on average $0.44\ \mathrm{m\,w.e.\,yr^{-1}}$ higher CMB in the AROME-ARCTIC-forced simulations. This is most likely due to a larger amount of precipitation and lower temperatures in AROME-ARCTIC than in CARRA. The average difference in glacier runoff between the AROME-ARCTIC and CARRA-forced simulations is -0.03 $\mathrm{m\,w.e.\,yr^{-1}}$, equivalent to only about 2% of the total runoff. Lower estimates of the runoff in the AROME-ARCTIC-forced simulations are found for Nordenskiöldland and Kvitøya. We, therefore, find that the AROME-ARCTIC forecast product provides a good estimate of the CMB and runoff overall, although the uncertainties have to be kept in mind for some areas, particularly Nordenskiöldland. We therefore suggest that AROME-ARCTIC forecasts could be used to generate continuously updating, high-quality simulations of the CMB, runoff and snow conditions in Svalbard. However, since this is an evolving product, technical difficulties may occur if data formats or naming conventions change, or if the forecast files are missing for longer than a few days. For many applications, however, using CARRA forcing may soon be enough, as it will in the future be updated on a monthly basis.

These CryoGrid simulations could be expanded to cover the whole CARRA-East and AROME-ARCTIC domains and thus provide valuable estimates of the runoff from all land areas in the Barents sea region. Knowledge of the climatic glacial mass balance and runoff from Franz Josef Land and Novaya Zemlya are sparse, and using the setup presented here could provide valuable insight. Since Svalbard, Franz Josef Land, and Novaya Zemlya experience similar climatic conditions, it is likely a model which performs well for Svalbard will also perform well for these regions. However, since less observational data are available for assimilation into the CARRA reanalysis and the AROME-ARCTIC forecasts, the uncertainties for these regions may be higher.

*Code and data availability.* The simulations described in this paper from both CARRA and AROME-ARCTIC forced simulations are available at

https://doi.org/10.21343/ncwc-s086 (Schmidt, 2022) at both a daily and monthly temporal resolution. They can be used for a wide range of applications, e.g. as input for runoff, ocean circulation or ecosystem models.

AWS data from MET-Norway is freely available from https://frost.met.no. The Kongsvegen AWS time series are also accessible at https://doi.org/10.21334/npolar.2017.5dc31930 (Kohler et al., 2017). Glacier-wide mass balances for Kongsvegen, Hansbreen, Holtedahlfonna, and Austre Brøggerbreen are available in the database of the World Glacier Monitoring Service (https://wgms.ch/).

AROME-ARCTIC can be downloaded from https://thredds.met.no/thredds/catalog/aromearcticarchive/catalog.html. CARRA data (Schyberg et al., 2020) was downloaded from the Copernicus Climate Change Service (C3S) Climate Data Store. The results contain modified Copernicus Climate Change Service information 2022. Neither the European Commission nor ECMWF is responsible for any use that may be made of the Copernicus information or data it contains.

The CryoGrid community model is hosted on Github. The source code is available at

https://github.com/CryoGrid/CryoGridCommunity_source

.

*Author contributions.* LSS and SW developed the model code. LSS performed the simulations, analyzed the results and produced the dataset and associated metadata. TVS helped with discussion and analyzing the results. EET provided CARRA inputs. LSS prepared the manuscript with contributions from all co-authors.

*Competing interests.* We declare no competing interests

*Acknowledgements.* We are grateful to Ward van Pelt, Maurice Van Tiggelen, and one anonymous reviewer for their detailed and constructive comments on the manuscript, which significantly improved this manuscript. We gratefully acknowledge the Carleen Tijm-Reijmer and the Institute for Marine and Atmospheric research Utrecht (IMAU) for providing AWS data from Nordenskiöldbreen and Ulvebreen. In addition, we acknowledge Øystein Godøy and Lara Ferrighi for their valuable help with data archieving. The AWS on Nordenskiöldbreen and Ulvebreen were funded by the Dutch Polar Programme of the Dutch Research Council (NWO-NPP). The research conducted in this study was funded by the Research Council of Norway through the Nansen Legacy project (NFR-276730). The simulations were performed on resources provided by the Department of Geosciences, University of Oslo.

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
