# Peer review of "Meltwater runoff and glacier mass balance in the high Arctic: 1991-2022 simulations for Svalbard"

_EGUsphere, 2022_

## Referee Comment (RC2)

[referee-annotated manuscript omitted]

---

## Referee Comment (RC3)

Review of "Meltwater runoff and glacier mass balance in the high Arctic: 1991-2022 simulations for Svalbard" by Louise Steffensen Schmidt et al.

This study presents a new dataset of modelled surface energy and mass balance fluxes over Svalbard using CryoGrid, and the new, high resolution reanalysis product CARRA as atmospheric forcing. The model is then extensively evaluated against a large database of weather station and mass balance observations, and finally used to quantify the changes in near-surface meteorology, glacier mass balance, and surface runoff over Svalbard during 1991-2022.

The methods used in this study are very suitable, and are also innovative in the sense that it is the first time that such a high-resolution atmospheric reanalysis product (CARRA 2.5km) is used to estimate the energy and mass balance using a surface/subsurface energy balance model for a period of 30 years. The evaluation of the new dataset is very detailed, and supports the overall capability of an energy balance model to simulate region-wide mass balance components. The significance of this work is large, since it describes a new modelling framework to estimate region-wide surface fluxes, where until now rather coarse and uncertain climate models are used for instead. Something that is not explicitly mentioned by the authors, yet increases the significance even more, is that such a database may serve as a unique benchmark dataset for climate models, since it contains many more variables than those available in an atmospheric reanalysis product like CARRA or ERA5. The paper is well written and well structured. The data is presented with clear figures and tables, but the manuscript is quite long (26 pages without counting the reference and the supplementary material).

In its present form the manuscript does not explicitly raise a research question, which I believe makes it harder for the reader to find the scientific significance of this work. Instead, this manuscript is a rather detailed description of a new, high quality dataset, and would therefore be very suited for a dataset journal such as ESSD.
Given the significance of this work for the community, I would nevertheless recommend publication in The Cryosphere, yet I would also recommend that both the abstract and the introduction are reformulated such that it becomes more clear what are the motivation and the scientific questions of this work.
In the conclusion it appears that an answer is given to these questions :

- How accurately can we model the energy and mass balance components using CARRA ?

- What is the mass balance of Svalbard glaciers and how did it change in the period 1991-2022 ?

- Can we realistically forecast in near-real time the surface fluxes in Svalbard using weather forecast model AROME-ARCTIC ?

Explicitly mentioning the research questions with a clear 'motivation $\rightarrow$ obstacle $\rightarrow$ solution' structure in the introduction would be very helpful. A clear obstacle I can think of, and that strongly motivates this work, is that climate models are possibly too coarse to accurately model the mass balance of Svalbard glaciers, and that statistical downscaling methods do not explicitly resolve the physical processes. In the next part some further suggestions are given with the aim to further improve this manuscript. Overall I would like to congratulate the authors on generating this dataset and on performing such a detailed analysis.

**Major comments**

- The introduction does a good job in describing the scientific literature, yet as mentionned above it could be more clear in describing the scientific challenge, and also more clear in explaining why using CryoGrid forced by CARRA offers a solution to the scientific challenge.

- It is not clear why AROME-ARCTIC is also used as a forcing for CryoGrid. It seems that CARRA and AROME-ARCTIC differ quite significantly, which is interesting yet not very surprising given the fact that CARRA assimilates some observations and that AROME is a weather forecast model (as mentioned by the authors in the conclusion as well). Regional difference between the two products may average out, but this still reduces the applicability/usefulness of a near-real time forecast of mass balance. Perhaps the authors could more clearly motivate the choice of using AROME-ARCTIC, or just briefly mention this possibly in the discussion and then remove section 6.2.

- How did the authors compute the temperature and relative humidity from automatic weather stations at 2 m, and wind speed at 10 m ? These variables are often measured at some height above the surface that changes between each maintenance, and also due to snowfall. Because of the large vertical gradients on a glacier, the deviation of T,WS and q in using a wrong sensor height can be quite large, which could make the evaluation with CARRA or ARMOE-ARCTIC inaccurate.

**Minor comments**

- Figure 1: Perhaps it would be useful for the non-expert reader to add the locations of each glacier/region mentioned in the text (Etonbreen, Austre Brøggerbreen, etc ...). It would also be very useful to add the coordinates and elevation of each AWS, or to refer to studies where this information is available.

- L69-71, It is not clear why the Russian Arctic is mentioned here since it does not re-appear in the methods or in the results. This interesting idea should be better mentioned in the discussion.

- Lines L83-84 can be made more consistent with L27-28.

- L186: Please clarify what modules of CryoGrid are part of Westermann et al (2022), and what modules have been specifically added for this study. Also, please explain how the 'water percolation and runoff modules' (4.2.1) differ from the original CryoGrid setup.

- L192: Why was a constant bare ice albedo of 0.4 chosen ? This seems very simplistic yet it appears that the errors in albedo are very small nonetheless (Table S2.).

- L205 I believe that this is commonly called a "bucket scheme". Then the variable $\theta_{fc}$ in Eq (1) is the irreducible water content (or " maximum liquid water-holding capacity " as defined in CROCUS by Vionnet et al 2012.), which is not necessarily the same as the field capacity (typically defined for soils, as far as I know).

- Also, if "Water is not allowed to flow into an impermeable layer, here defined as layers with a density higher than 830 kgm3" (L219), it is interesting and perhaps counter-intuitive to see in the results that "the average annual internal accumulation is 0.11 mw.e., and thus accounts for almost half of the total refreezing (Fig. 9c)". Perhaps the authors could comment on this in the discussion ?

- L209: Consider replacing "water saturation" by " volumetric water content in the snow" to be consistent with other snow models.

- L225 How is the "water in excess of the field capacity" defined ?

- L256-259. I acknowledge that the manuscript will be a bit long if the AWS evaluation is further detailed here. Yet I find the statement "both models generally fit well with observations" a bit simplistic, not very specific and possibly also inaccurate. It would be very useful for future studies to better understand what causes the limitations of a SEB model forced by CARRA. For instance it seems that summer ablation is overestimated at Nordenskiöldbreen (figure 4, red lines), which could partly be explained by an underestimated albedo by 0.06 (Table S1). Also, it appears that there is a systematic underestimation in both incoming and outgoing longwave radiation components at all glacier AWS.

**Technical comments**

- L10 "an"

**References**

Vionnet V, Brun E, Morin S, et al (2012) The detailed snowpack scheme Crocus and its implementation in SURFEX v7.2. Geosci Model Dev 5:773–791. https://doi.org/10.5194/gmd-5-773-2012

---

## Author Response (AR1)

**Reviewer 1**

We are thankful to the reviewer for taking the time to review this manuscript. We have answered all the comments of the reviewer below.

The reviewer comments appear in bold, our answers in normal font, and changes to the manuscript in italics.

**I suggest then to the authors to resubmit their paper in GMD (Geoscientific Model Development) which fits a lot of better with this kind of papers.**

We disagree, this is not purely a model description paper. A large paper describing the CryoGrid model is already available in GMD (Westermann et al, 2022), and describes the model in much deeper detail than we do here. A journal like Earth System Science Data could have been fitting, but they do not allow model output descriptions from climate model/land surface models. At the suggestion of another reviewer, we have now restructured the introduction to make the research aims clearer.

**Before resubmission, I would just like to highlight that I don't see the interest of the addition of AROME-ARCTIC as forcing of the snow model in this paper. With respect to CARRA, AROME-ARCTIC is an operational product forced by IFS and can not be used in any scientific relevant papers knowing that ERA5 is available in real time with a delay of 1 week with the present date and only using AROME-ARCTIC forced by ERA5 will be relevant here for me. If the aim of the authors is to show the interest of AROME-ARCTIC used in forecast mode, I suggest in this case to change the focus of the paper to this aim by considering only 2016-2021. But, discussing the recent changes over 1991-2021 + evaluating the interest of AROME-ARCTIC over 2016-2021 in the same time decreases the interest of the paper for me.**

We disagree that AROME-ARCTIC is not useful because ERA5 exists. ERA5 is a useful product, but the resolution is too low for simulations of glaciers on Svalbard (30 km vs the 2.5 km of AROME-ARCTIC). AROME-ARCTIC uses the same model as CARRA with a very similar setup and assimilates most of the same data. It therefore makes sense to evaluate the two products in the same paper, particularly since AROME-ARCTIC simulations have already been used in previous studies. However, we try to more clearly separate the results based on CARRA from those based on AROME-Arctic, such that readers who are not interested in AROME-ARCTIC, can simply skip section 5.4 of the results.

**Some additional minor remarks:**

**The use of the words "climatic mass balance" or "glacier mass balance" is ambiguous here for me as both mean surface mass balance. I suggest than to use SMB everywhere in the paper.**

There is a difference between the total mass balance, climatic mass balance, and surface mass balance, and we therefore would like to keep the distinction. The surface mass balance (SMB) quantifies the mass fluxes between the atmosphere and the glacier at the surface, and within the current year's snow layer (refreezing within the annual layer). This is what is measured by in-situ glaciological observations. The climatic mass balance (CMB) additionally accounts for mass changes below the last summer surface and can therefore be simulated with a model like the one used in this study. Total glacier mass balance is the sum of CMB, basal mass balance and frontal ablation (i.e., subaqueous melting and calving). The latter

term only applies to glaciers terminating in the ocean or freshwater lakes. The terminology used in our manuscript follows that recommended by Cogley et al (2011)

We have made this distinction clearer in the introduction:

*L80-82: CryoGrid simulates both the surface mass balance (SMB) and the climatic mass balance (CMB). The surface mass balance quantifies the mass fluxes between the atmosphere and the glacier at the surface, as well as refreezing within the annual layer. The SMB is what is measured by in-situ glaciological observations. The climatic mass balance additionally accounts for mass changes below the last summer surface. For tidewater glaciers, CryoGrid, however, cannot calculate the total glacier mass balance, as this is the sum of CMB, basal mass balance and frontal ablation, and the latter cannot be determined from an energy-balance model.*

**In Section 5.6, runoff is given in GT/yr while SMB/precip is discussed in mWE/yr. I suggest to use the same units (m WE/yr or GT/yr) through all the paper.**

This is a good point. We have given the runoff in Gt/yr, as this unit is often used for runoff, while m w.e./yr is often used for the SMB. However, in order to better compare, we have now added the runoff given as m w.e./yr in both the runoff figure and in the text, so it is easier to compare. We still, in some cases, give the runoff in Gt/yr in addition so it easy to compare with the runoff from other studies.

**What is the interest of Section 5.4? It is mainly a comparison between CARRA and AROME-ARCTIC forcing but what are impacts of the seasonal snowpack recent changes on SMB or climate? A comparison with observations could be useful if the aim is to validate it.**

The idea was to show how the seasonal snowpack has evolved over the last 30 years, but you are right that it is maybe not the most relevant for this paper. We have removed the section from the revised manuscript.

**In conclusion, increases in land runoff is mentioned without having discussed more in depth changes in precipitation (rainfall/snowfall). Discussing land vs glacier trend will be useful here in addition to use the same units for precipitation and runoff. More in general, what is the interest of discussing here the land runoff changes? What are the impacts on ocean? Are there some observations confirming this modelled estimate?**

We have now added a bit more discussion on the changes in precipitation (rainfall vs snow):

*L290-298: The average precipitation over Svalbard is 0.62 m w.e. yr−1 . There is a small but significant trend in the average yearly precipitation of 0.05 m w.e. decade−1 (p < 0.01). There is a larger trend in the precipitation over glacier-covered points (0.06m w.e. decade−1) than non-glacier-covered points (0.03m w.e. decade−1). Although there is no significant trend for all areas of Svalbard, there is a positive trend over e.g. Austfonna, Vestfonna, and North Spitsbergen. The largest trend is over NE Austfonna of 0.17 m w.e. decade−1. Over the investigated period, on average 90% of the precipitation fell as snowfall. There is a significant decreasing trend in the ratio between snow and rain, with the percentage of precipitation falling as snow decreasing by -2.0% decade−1 (p=0.01). For glacier-covered points, 95% of the precipitation falls as snow, with a significant decreasing trend of -1.3% decade−1 (p=0.02). Over non glacier-covered points, however, 85% of the precipitation falls as snow, with a significant decreasing trend of -2.8% decade−1 (p=0.01).*

The interest in discussing land runoff is to compare it to the runoff from glaciers to get an idea about the runoff contributions from seasonal snow. Runoff from seasonal snow generally occurs earlier in the year, so it therefore affects the runoff into the ocean. In addition, the effect of the ocean is different depending on if the runoff comes from a tidewater glacier or from seasonal snow or land terminating glacier (e.g. in terms of nutrients, light pollution, fjord circulation). This is, however, not the focus of this study. But we have now added a few lines in the conclusion to highlight the significance of runoff for the ocean:

*L558-59: The timing and amount of freshwater runoff from Svalbard has important implications for the ecosystems in the surrounding fjords. Changes is freshwater discharge affect a wide range of physical, chemical, and biological processes, including e.g. fjord circulation (Carroll et al. (2017)), light availability (Hop et al. (2002); ARIMITSU et al. (2012)), water biogeochemisty (Wadham et al. (2013); Bhatia et al. (2013)), and marine primary production (Juul-Pedersen et al. (2015); Hopwood et al. (2020)). Freshwater from tidewater glaciers may affect these processes in a different manner from seasonal snow runoff or runoff from land-terminating glaciers, and it is therefore important to quantify the amount of different types of runoff.*

We had not included any evaluation of the runoff, as very few observations are available from glaciated catchments. Now we have included a comparison with two partially glaciated catchments (Bayelva and De Geerdalen) as section 5.2.3.

Citations:

Cogley, J.G., R. Hock, L.A. Rasmussen, A.A. Arendt, A. Bauder, R.J. Braithwaite, P. Jansson, G. Kaser, M. Möller, L. Nicholson and M. Zemp, 2011, Glossary of Glacier Mass Balance and Related Terms, IHP-VII Technical Documents in Hydrology No. 86, IACS Contribution No. 2, UNESCO-IHP, Paris.

**Reviewer 2:**

We are very grateful for the constructive comments and suggestions provided by the reviewer that have significantly improved our manuscript. We agree that including the entire Barents sea area would have been interesting, but since the paper was already very long, we have saved that comparison for another manuscript. We have included all suggestions for changes and have outlined the major comments below.

The reviewer comments appear in bold, our answers in normal font, and changes to the manuscript in italics.

**L9: Might be better to use CMB instead as capital letters are more easy to recognize as an abbreviation**

That is a good point, we have changed this in the text and figures.

**L14: It should not be forgotten that CARRA has its uncertainties as well, probably of similar magnitude AROME-ARCTIC, as both are using some climate model constrained with (largely the same) sets of observational data. Just saying that small differences between the two datasets do not necessarily mean good performance of one of the two**

Good point, we have made it clearer in that this only means they will provide similar quality predictions:

*L17-19: This indicates that AROME-ARCTIC may provide similar high-quality predictions of the total mass balance of Svalbard as CARRA, but regional uncertainties should be taken into consideration.*

**L68: Why is this mentioned here? It almost gives me the impression that the Russian Arctic mass balance will be presented as well in this study, but that does not seem to be the case.**

The idea was to first test the model for Svalbard, where many observations are available, and show that the model setup works well in the Barents Sea area. But you are right, it is misleading, and we have removed the section now.

**L89: It could be good to modify figure 1 so that it also includes geographic names that are used in this study (oceans, places and regions in Svalbard etc). For example, many readers will not know the difference between Spitsbergen and Svalbard.**

Good point, geographic names of oceans, places and regions used have been added to Figure 1.

**L104: Could it be summarized here in a few sentences?**

We have added the following description:

*First, both the CARRA reanalysis and AROME-ARCTIC forecasts are evaluated against available observations from automatic weather stations (AWSs). Unsurprisingly, both products performed well when compared to AWS data which had been assimilated into the forcing products but had larger biases when compared to glacier measurements which had not been assimilated. The comparison of AROME-ARCTIC and CARRA at the AWS locations show that both products were similar, albeit with larger biases and root-mean-square-errors for AROME-ARCTIC. In addition, the consistency between the two forcings is evaluated for the overlap period (2016-2021). We found that AROME-ARCTIC is on average colder than*

*CARRA, particularly in NW Spitsbergen where the average yearly temperature was -2◦C colder in AROME-ARCTIC. The full results of this analysis are described in Supplement S2.*

**L105: It could be worth mentioning the maximum depth of the subsurface model and the vertical resolution already here.**

The following sentence has been added here to the text:

*L119-20: The model is initialised with 47 layers of ice with a thickness between 0.1 and 1 m, totalling 20 m of glacier ice.*

**L118: I support this approach. Still, it is good to realize (and possibly discuss) that using the same initial conditions will probably reduce differences in calculated mass balance with both forcings**

This is a good point; it reduces the differences (which is also one of the reasons we use the same initialisation). We have added the following lines to make this clearer to the reader:

*L133-34: This most likely will reduce the difference in CMB calculated using the two products, compared to if different spin-ups were produced.*

**L126: Could be good to specify the "final results" here. I suppose runoff and snow depth can simply be weighted. But what is done with glacier-specific variables like cmb?**

We do not add the land and glacier components together (although it could be done for runoff like you said). What is meant is that to calculate the average or sum of a variable for glaciers, we do a weighted average/sum based on the glaciated area in a point. Similar for non-glaciated points, when we are calculating the total value for all of Svalbard, a weighted average or sum is done based on the area-fraction. We have clarified this in the text as:

*To calculate the average or sum of a variable for a specific region or all of Svalbard, the results are weighted based on the fractional glacier coverage.*

**Figure 1: It would be nice with a bit larger and detailed map, with some place names. Also elevation contours could be useful**

The figure has been made larger and elevation contours and place names mentioned in the text have been added.

***L158: Maybe it could be explained in this paragraph how 2.5x2.5 km results are compared to point observations. What was done when several mass balance observation points fell in one grid cell of the model?***

The following sentence has been added to explain this:

*L175-77: When several observation points fall within one 2.5 x 2.5 km model grid, only the measurement point closest to the center of the gridpoint is used.*

**L183: Has CryoGrid been calibrated in any way? Have observations been used to optimize uncertain parameters in the energy balance and snow routines? In case this is described in Westermann et al. (2022) please indicate that here.**

The only calibration that has been done against observations is using the mass balance and albedo observations. Most values are taken from other studies, and not many variables needed to be altered from the standard (we e.g. tested the snow/rain temperature threshold, but the best results were found using the CryoGrid default of 2 degrees). We did calibrate the ice albedo and found that 0.4 provided the best results. This has been added to the text:

*L222-24: Previous mass balance studies of Svalbard have used ice albedo values in the range of 0.3-0.4 (Østby et al, 2017; van Pelt et al, 2019) for all of Svalbard. From calibration with available mass balance and albedo observations, we found the best results using an ice albedo of 0.4.*

**L233: any reason for using snow water equivalent rather than snow thickness as a threshold? I am just curious because it is mostly the (minimum) snow depth of layers that affects numerical stability, e.g. of the heat diffusion equation, rather than minimum snow mass.**

The stability is affected by both the minimum snow depth and the heat capacity: if the heat capacity is low, the same energy input will lead to a stronger temperature change which leads to instabilities. Using the SWE as a threshold captures both these sources of instability – it keeps the grid cells from getting too thin, and it keeps the heat capacity from fluctuating.

**L237: How is the fresh snow density set? Is it constant or variable?**

The fresh snow density is variable and depends on the air temperature and wind speed. The equation is given in the supplemental material (Eq S5).

**L278: Nordenskiöldbreen is tricky! Especially because of high wind speeds and snow drift at low elevations, whereas higher elevations have much calmer conditions. This generates a very strong accumulation - elevation gradient.**

Good point! We have added a description of this to explain the higher biases for this glacier.

**L293: And over what periods are summer and winter balance defined? Maybe I missed it...**

The summer balance is defined as from April 1- August 31, and the winter balance is from September 1$^{st}$ – March 31st. We have added this to the text:

*L351-52: For calculations of the winter and summer mass balance, we use fixed dates of 1 April and 1 September.*

**Figure 6: It is interesting that AROME-ARCTIC simultaneously has a more positive summer balance and a more negative winter balance. It would have been more likely that a negative winter balance anomaly would also give a negative summer balance anomaly since less snow in winter usually means more melt in summer. Why is this not the case here? Maybe other weather variables, e.g. cloud cover, provide an explanation? It would be interesting to add a brief discussion on this in the manuscript.**

This is indeed different than what you normally expect. AROME-ARCTIC generally has lower temperatures than CARRA, also in the summer, which of course leads to less melting. In addition, although the precipitation over the year is very similar in the two products, there is a bit less precipitation in the winter in AROME-ARCTIC and a bit more in the summer. This is currently discussed in the Supplement S2. A combination of lower temperatures and slightly different precipitation patterns is probably what causes the underestimation of both the summer and the winter balance.

**L314: Please be consistent with the units (m w.e. or m w.e. yr-1)**

We have changed the units to always be m w.e. yr-1

**L328: Was this always the case? I suppose in cold years this may not be true everywhere. I can also imagine that there may be points in Svalbard for which the model simulates a positive mass balance, but which are not part of a glacier (just because of model uncertainty). It could be good to mention somewhere how such points were dealt with (in case they occurred) when calculating average snow onset / disapearance dates**

This section was removed at the request of another reviewer. But this is true, these points did occur – both due to model uncertainty and during cold years. This is only an issue during a few years, and on average only affects 4% of the land grid points. However, in extreme years this can be up to 30%. To avoid this, the model removes all perennial snow on September 1st, and thus for the calculation of snow disappearance day, this value is used.

**L356: Could it also play a role that the AROME-ARCTIC simulation (presumably) starts from subsurface conditions that were initialized with the CARRA forcing?**

Yes, this could definitely also be a factor.

**L364: The fact that there are no points with 0 runoff may imply that there is no deep firn in the model results that is still cold (<0 deg C). Is that indeed the case?**

Yes, this is correct. The large melt years of 2013 and 2020 caused the deep firn to heat up and become temperate. This is also partly related to the spin-up – the period we use for spin-up is relatively warm, which means that the firn likely is too warm in the beginning of our simulations, and thus heat up too fast.

**L367: The time-series in Fig. 10 seem to agree really well with what was presented in Van Pelt et al. (2019) in a similar figure. Both the absolute values and year-to-year variations match very well.**

This is true, they do match up very well! We also mention this in section 6.4.

**L392: Right now only a range in calving rates is given for 2005-2009. To be consistent it would be good to indicate a similar range for all presented periods. Furthermore, it could be mentioned that high calving since 2010 is likely the result of the surge of Basin-3.**

We have added the ranges for all presented periods, and have added the following sentence about Basin-3:

*L456-57: The calving after 2010 is likely increased due to the surge of Basin-3, the largest outlet basin of the Austfonna ice cap, which significantly increased the calving from the ice cap (e.g. Dunse et al, 2015).*

**L394: Please note that also CARRA reanalysis data will from 2023 onwards be updated on a monthly basis (https://climate.copernicus.eu/copernicus-arctic-regional-reanalysis-service). That could be sufficiently "real-time" for many applications.**

This is true, we have added a line mentioning this in the conclusion:

*L575-76: For many applications, however, using CARRA forcing may soon be enough, as it will in the future be updated on a monthly basis.*

**L400-408: I am not fully sure what the main point here is. Maybe it could be clarified. It does nicely show, including error bars, that 2021/22 was indeed the most negative mass balance year since (at least) 1991. Maybe that could be highlighted.**

We have now removed this section, and instead it is a part of a specific results section on AROME-ARCTIC (section 5.4). We now highlight that 2021/22 was the most negative mass balance year more clearly in the text.

*L442-44: Based on AROME-ARCTIC, 2021/22 is a record negative mass balance year for Svalbard, with a CMB of -0.86 m w.e. yr−1. There is a highly negative mass balance in all regions in Svalbard. The runoff from glaciers is also the highest over the simulation period, of 1.6 m w.e. yr−1 (58Gt yr−1)*

**L423: Does this apply to only the snow and firn parameters or also the energy balance parameters?**

This is referring to the snow/firn parameters in the Vionnet/Royet paper. We have clarified this in the text:

*L468-69: However, most of the model parameters used by the snow and firn scheme are based on recommendations from previous studies and have not been tuned for the conditions of Svalbard.*

**L461: Please note that the effects of ignoring elevation and mask change were also investigated for a future projection for Svalbard in Van Pelt et al. (2021; https://doi.org/10.1017/jog.2021.2). See Fig. 13 and the related discussion in that study. It is found that the elevation and outline changes have counteracting mass balance effects that are approximately in balance.**

Thanks for pointing this out, it very relevant for the discussion. We have added the following to the text:

*L527-30: In addition, van Pelt et al (2021) investigated the effect of ignoring both elevation and glacier mask changes on future projections for Svalbard from 2018-2060. Over this time period, the authors found that the increased melt due to a lowering of the glacier surface was nearly balanced by the melt reduction due to a changing glacier mask, and thus the introduced error in the runoff and CMB was small.*

**Reviewer 3:**

We are very grateful for the constructive comments and suggestions provided by the reviewer that have significantly improved our manuscript. We agree that the manuscript would also have fitted well within a journal like ESSD, but unfortunately they do not publish outputs from climate models or land surface models. We have restructured the introduction based on your helpful suggestions, and the research aim of the paper is now hopefully clear.

In general, we have included all suggestions for changes and have outlined our answers to all comments below. The reviewer comments appear in bold, our answers in normal font, and changes to the manuscript in italics.

**Major comments**
**The introduction does a good job in describing the scientific literature, yet as mentionned above it could be more clear in describing the scientific challenge, and also more clear in explaining why using CryoGrid forced by CARRA offers a solution to the scientific challenge.**

Thank you very much for your helpful comment! We agree that the scientific challenges should be mentioned more clearly. We have now restructured the introduction following your above suggestions as:

- Motivation: accurate estimates of the past evolution of mass balance and runoff
- Obstacle: global reanalysis products are too coarse, and statistical downscaling does not resolve all physical processes
- Solution: regional high-analysis productions which assimilate local observations, such as CARRA or AROME-ARCTIC. We evaluate these products for use in mass balance simulations and use the output to investigate the climatic changes in Svalbard over the last 3 decades.

**It is not clear why AROME-ARCTIC is also used as a forcing for CryoGrid. It seems that CARRA and AROME-ARCTIC differ quite significantly, which is interesting yet not very surprising given the fact that CARRA assimilates some observations and that AROME is a weather forecast model (as mentioned by the authors in the conclusion as well). Regional difference between the two products may average out, but this still reduces the applicability/usefulness of a near-real time forecast of mass balance. Perhaps the authors could more clearly motivate the choice of using AROME-ARCTIC, or just briefly mention this possibly in the discussion and then remove section 6.2.**

We have tried to motivate this choice more in the introduction:

*L69-72: In addition, we investigate if the forecast product AROME-ARCTIC, which uses the same model and similar observations as CARRA, can be used to extend the CARRA product, thus providing almost real-time updates of the mass balance and runoff. Almost real time-simulations could provide valuable information for e.g. fieldwork planning (to check the current conditions) and public outreach.*

In addition, we have tried to have less focus on AROME-ARCTIC in the paper and more on the CARRA simulations. The AROME-ARCTIC simulations are now mostly described in a separate section in the results section (Section 5.4), and section 6.2 has been removed.

**How did the authors compute the temperature and relative humidity from automatic weather stations at 2 m, and wind speed at 10 m ? These variables are often measured at some height above the surface that changes between each maintenance, and also due to snowfall. Because of the large vertical gradients on a glacier, the deviation of T,WS and q in using a wrong sensor height can be quite large, which could make the evaluation with CARRA or ARMOE-ARCTIC inaccurate.**

AWS on glaciers are set-up as so-called floating stations, i.e. they are based on a tripod standing on the ice surface, rather than fixed to a mast drilled into the ice. That way, sensors keep constant height above the ice surface during the snow-free period, as the station moves together with the ice surface during melting. During the snow cover period, the height of sensors is reduced by the snow height. Snow depths on Svalbard are modest and seldom amount to more than 1 m at AWS sites. We have not attempted to correct AWS data for this effect for two reasons: i) the largest effects of incorrect T,WS and q are on the calculation of turbulent fluxes which in turn have moderate significance for melting that predominantly occurs when the sensor height problem is smallest/ does not exist; ii) variations in sensor height due to snow cover do equally occur at AWS outside glaciers; where this typically is not either taken into account.

**Minor comments**
**Figure 1: Perhaps it would be useful for the non-expert reader to add the locations of each glacier/region mentioned in the text (Etonbreen, Austre Brøggerbreen, etc ...). It would also be very useful to add the coordinates and elevation of each AWS, or to refer to studies where this information is available.**

This is a good point; we have names of regions and glaciers mentioned in the text. We have also added the coordinates and elevations of each AWS in Table 1.  Studies where this information is available are also mentioned in the text.

**L69-71, It is not clear why the Russian Arctic is mentioned here since it does not re-appearin the methods or in the results. This interesting idea should be better mentioned in the discussion.**

This has been deleted from the introduction and is instead only mentioned at the end of the conclusion as an outlook to where this method could be used in the future.

**Lines L83-84 can be made more consistent with L27-28.**

The sentence has been changed to be more consistent with L27-28:

*L91-93: Located in the Norwegian Arctic between 75 and 81°N, the Svalbard archipelago is in one of the currently fastest warming regions in the world, the Barents Sea region (Screen and Simmonds, 2010; Lind et al 2018}, and has the strongest observed warming in Europe since the 1960's (Nordli et al, 2014),*

**L186: Please clarify what modules of CryoGrid are part of Westermann et al (2022), and what modules**

**have been specifically added for this study. Also, please explain how the 'water percolation and runoff modules' (4.2.1) differ from the original CryoGrid setup.**

We have now made this clearer in the text in several locations:

*L220-21: The glacier module consists of layers of pure-ice with a user-defined constant ice thickness. This module has not been altered compared to the one described in Westermann et al (2022)*

*---*

L231-38: *These modules have been specifically added to the model for this study.*

*The snow and firn modules follow a slightly altered CROCUS (Vionnet et al, 2012) snow scheme as described in Westermann et al (2023). Some of the main differences to the snow schemes presented in Westermann et al (2023) are:*

- *Additional output variables, including refreezing, internal accumulation, CMB, SMB.*
- *Updated water percolation and runoff scheme, including a parameterisation for the hydraulic conductivity and a runoff timescale (described in Section 4.2.1)*
- *Regridding of layers below the surface (described in Section 4.2.2)*

*---*

*L246-47: In Westermann et al (2023), a constant user-defined hydraulic conductivity is used. Here, the hydraulic conductivity is parameterized in terms of [...]*

**L192: Why was a constant bare ice albedo of 0.4 chosen ? This seems very simplistic yet it appears that the errors in albedo are very small nonetheless (Table S2.).**

We decided to use an ice albedo of 0.4, as this gave the best results when calibrated against observations of mass balance/albedo. We have now added a sentence about the calibration:

*L222-24: Previous mass balance studies of Svalbard have used ice albedo values in the range of 0.3-0.4 (Østby et al, 2017; van Pelt et al, 2019) for all of Svalbard. From calibration with available mass balance and albedo observations, we found the best results using an ice albedo of 0.4.*

Previous studies have used a constant ice albedo ranging from 0.3 to 0.4 (e.g. Østby et al, 2017; van Pelt et al, 2019) with good results, but this is of course a simplification, as the ice albedo varies from 0.15 to 0.44 across Svalbard (Greuell et al, 2007). Using albedo observations, e.g., from MODIS, to create an ice albedo map would probably give better results and could be included in the model in the future. We have added a few sentences on this in the discussion:

*L471-74: In addition, we use a constant ice albedo in the model, which could be a major simplification given that the ice albedo varies across Svalbard from 0.15 to 0.44 (Greuell et al, 2007). In future work, this could be improved by using estimates of the ice albedo from e.g. MODIS observations to create a map of the ice albedo (Schmidt et al, 2017) and/or updating the albedo parameterisation to account for dust and impurity content.*

**L205 I believe that this is commonly called a "bucket scheme". Then the variable $\theta_f$ c in Eq (1) is the**

**irreducible water content (or " maximum liquid water-holding capacity " as definedin CROCUS by Vionnet et al 2012.), which is not necessarily the same as the field capacity(typically defined for soils, as far as I know).**

Thank you for noticing this, we have changed "field capacity" to "irreducible water content"

---

## Author Response (AR2)

**Reviewer 1 – round 2**

We thank the reviewer for their careful reading of the manuscript, and for noticing the problem with the different units. We have answered all comments below. The reviewer comments appear in bold, our answers in normal font, and changes to the manuscript in italics.

**- the problem of using different units (m W.E or GT) remains. For example, in the abstract (lines 10-15, in the authors_tracked_changes_file) SMB is given in mWE, runoff in GT and refreezing in m WE. All the SMB components should be listed in the same units as SMB is the sum of them. We usually use GT as unit for integrated value of SMB and TMB but I'm OK if the authors prefer to use m WE/yr bt in this case, it should be also the case for runoff.**

Thanks for noticing this, you are right that we did not change the units in the abstract and conclusion. This has now been fixed.

**- line 109: what if the difference between CMB (climatic mass balance), total mass balance and SMB (surface mass balance). It is the 1st time that I see a mention of CMB. We usually use total mass balance (TMB) for SMB + ice dynamics mass balance.**

As mentioned in the first review, we are following the recommended terminology of climatic mass balance, surface mass balance, and total mass balance as given by Cogley et al (2011). Previous papers on Svalbard have made a similar distinction (e.g. Østby et al (2017), van Pelt et al (2019 and 2021), Schuler et al (2020)).

The surface mass balance (SMB) quantifies the mass fluxes between the atmosphere and the glacier at the surface, and within the current year's snow layer (refreezing within the annual layer). This is what is measured by in-situ glaciological observations.

The climatic mass balance (CMB) additionally accounts for mass changes below the last summer surface (in the firn layer) and can therefore be simulated with a model like the one used in this study.

Total glacier mass balance is the sum of CMB, basal mass balance and frontal ablation (i.e., subaqueous melting and calving).

We had already described this difference in the introduction in the revised manuscript:

*L82-88: CryoGrid simulates both the surface mass balance (SMB) and the climatic mass balance (CMB). The surface mass balance quantifies the mass fluxes between the atmosphere and the glacier at the surface, as well as refreezing within the annual layer. The SMB is what is measured by in-situ glaciological observations. The climatic mass balance additionally accounts for mass changes below the last summer surface. For tidewater glaciers, CryoGrid, however, cannot calculate the total glacier mass balance, as this is the sum of CMB, basal mass balance and frontal ablation, and the latter cannot be determined from an energy-balance model.*

But have tried to clarify further by changing the text slightly to the following:

*L82-88: CryoGrid simulates both the surface mass balance (SMB) and the climatic mass balance (CMB). The SMB quantifies the mass fluxes between the atmosphere and the glacier at the surface, as well as*

*refreezing within the annual layer. The SMB is what is measured by in-situ glaciological observations. The CMB additionally accounts for mass changes below the last summer surface, e.g. in the deeper firn layers. The total mass balance, the sum of CMB, basal mass balance and frontal ablation, cannot be calculated for tidewater glaciers by an energy-balance model like CryoGrid, as glacier dynamics are not included. This terminology follows that suggested by Cogley et al. (2011).*

**- line 214, for future validation, when several observation points fall within one 2.5 x 2.5 km model grid, I suggest rather to use the one the closest in altitude of the model or the average of all of them.**

Thanks for the suggestion, using the altitude is indeed better, particularly when there are differences between the modeled and measured altitude. In these cases, however, the point closest to the midpoint did correspond well to the model elevation.

**- line 312: what is the value used for the irreducible water content Wex?**

The irreducible water saturation is 0.05, following Vionnet et al (2012), and Wex is thus 0.05*porespace. We have clarified this in the text:

L294-295: *The irreducible water saturation is 0.05, following Vionnet et al. (2012), and the irreducible water content is thus 5% of the total porespace.*

**- Fig 9 showing times series in m WE and in GT is not relevant here. Mapped value should be in mmWE/yr or mWE/yr and integrated value in GT/yr or everything in mWE/yr**
We think it is relevant to also show the integrated values in both GT/yr and mWE/yr - in mWE/yr to be consistent with the CMB and refreezing, as you have previously pointed out, and in GT/yr so it is easier to compare with previous studies which provides similar figures in Gt/yr (e.g. van pelt et al, 2019)

**- in Table3 adding the 2016-2021 statistics of both CARRA and AROME models will be relevant.**
Table 3 shows comparison to previous studies, which at maximum includes results until 2018. Comparing to CARRA and AROME simulations would only be for 2016-2018, and not add much to the comparison. If the reviewer means table S3, we already have the statistics for both models in table S4.

**- lines 1673-283: there is the same problem with the units. SMB is given in mWE and runoff in GT/yr**
Thank you for noticing this, we have changed the runoff to mWE/yr.

**- line 700: this sentence should be in the "data availability" section (line 709**

The sentence has been moved as suggested

**Reviewer 3 – round 2**

We thank the reviewer for reading the manuscript again and their helpful comments. Sorry that we missed some of the comments last round, we have now answered the ones we missed.

The reviewer comments appear in bold, our answers in normal font, and changes to the manuscript in italics.

**The authors have addressed all my three major comments from the previous round of revisions, but have not responded to the last 4 (out of 10) minor comments. These comments are meant to better understand the methods and possibly provide the interested reader with some additional discussion points. In particular I believe it would be very helpful for future work to further quantify and discuss the errors of the model in part 5.2.1, especially since the main goal of this research is to have accurate estimates of surface mass balance.**

I am sorry; I must somehow have missed the last of your minor comments when I copied everything into a word document for the replies. We have answered all the comments below, including those we missed last round:

**Also, if "Water is not allowed to flow into an impermeable layer, here defined as layers with a density higher than 830 kgm3" (L219), it is interesting and perhaps counter-intuitive to see in the results that "the average annual internal accumulation is 0.11 mw.e., and thus accounts for almost half of the total refreezing (Fig. 9c)". Perhaps the authors could comment on this in the discussion ?**

Due to the re-gridding of model layers, it is quite rare that an impermeable layer occurs high up in the snow pack (or if it does, it often disappears as layers are merged). Impermeable layers mostly occur in the deep firn in the model, and thus refreezing can occur below the yearly layer, leading to a high internal accumulation. For model simulations where more thin layers are used than here, we would allow water to penetrate thin layers as it is unlikely that e.g. a 1 mm layer would completely stop water percolation.

**L209: Consider replacing "water saturation" by " volumetric water content in the snow" to be consistent with other snow models.**

Changed

**L225 How is the "water in excess of the field capacity" defined ?**

The irreducible water saturation is set to 0.05, following Vionnet et al (2012). The water in excess of the field capacity is thus any water quantity exceeding 0.05*total_porespace. We have now added the following sentence to clarify this:

*L 294-295: The irreducible water saturation is 0.05, following Vionnet et al. (2012), and the irreducible water content is thus 5% of the total porespace.*

**L256-259. I acknowledge that the manuscript will be a bit long if the AWS evaluation is further detailed here. Yet I find the statement "both models generally fit well with observations" a bit simplistic, not very specific and possibly also inaccurate. It would be very useful for future studies to better understand what causes the limitations of a SEB model forced by CARRA. For instance it seems**

**that summer ablation is overestimated at Nordenski¨oldbreen (figure 4, red lines), which could partly be explained by an underestimated albedo by 0.06 (Table S1). Also, it appears that there is a systematic underestimation in both incoming and outgoing longwave radiation components at all glacier AWS**

We have now added the following text about the evaluation of the CARRA forcing:

*L329-344: The comparison of the CARRA forcing against observations from automatic weather stations shows a general good agreement. The MET Norway stations have been assimilated into the CARRA product, and it is therefore not surprising that there is a good agreement between the two. The largest differences in temperature are found for the Sveagruve II station (ΔT = −1.8C), but for most of the MET Norway stations the mean temperature difference is below 1C. The largest differences in relative humidity and wind speed are found at Kvitøya (ΔRH = 6.4%) and Pyramiden (ΔWS = −1.9 m s−1), respectively.*

*The 2m temperature at the glacier stations, which were not assimilated into the CARRA product, is generally well represented, with biases generally smaller than 1◦C. The exception is at the Etonbreen AWS, where CARRA has a cold bias. This can, however, partly be attributed to a warm bias in the AWS observations over time at this station due to sensor drift, before redundancy has been installed in 2016. The relative humidity has a maximum bias of 6.2%, while the wind speed bias ranges between -1.3 and 1.5m s−1. The incoming longwave and shortwave radiation in CARRA generally fits well with the observations, albeit with a small negative bias in the longwave radiation for most of the stations (ranging between -1.6 and -14W m−2).*

*The evaluation of both forcing products against available AWS observations shows that the two products often provide similar results, but that the bias and root-mean-square-error of the CARRA product is generally smaller than for AROME-ARCTIC. For detailed evaluation of the model forcing against available AWS observations for both CARRA and AROME- ARCTIC, in addition to a discussion on the inter-comparison, we refer to Supplement S2, S3.*

Regarding Nordenskiöldbreen, this is generally tricky to simulate. This is especially because of high wind speeds and snow drift at low elevations, whereas higher elevations have much calmer conditions. This created a very strong accumulation – elevation gradient, which affects e.g. the albedo and thus the ablation. We added the following lines in the manuscript describing this during the last review round:

*L366-368: Nordenskiöldbreen experiences a very strong accumulation-elevation gradient, due to high wind speeds and snow drift at lower elevations and calmer conditions at higher elevations. It is therefore difficult to accurately simulate this glacier without including snow re-distribution between grid points.*

**Regarding my 2nd major comment; it would still be useful for future evaluations, and technically more correct, to mention at L209-210 from the tracked changes document that the temperature and wind speed are not always measured at 2m and 10m respectively, especially not on the glaciers. For the IMAU AWS at Nordenskiöldbreen and Ulvebreen the sensor boom is located between 3m and 4m depending on the tripod design. The errors in observed q, T and WS should not affect the modelled turbulent fluxes since CARRA is used as forcing. The errors will however affect the statistics of the evaluation presented in the supplementary material.**

You are right that we should mention how we deal with different sensor heights. We have now mentioned how we handle that the measurements are not at exactly two or ten meters in the text. For windspeed, we assume neutral stratification and a roughness length of 1 mm. For temperature and relative humidity, we use the temperature gradient between two model layers.:

*L176-182: When available, daily mean observations of the 2m temperature, 2m relative humidity, 10m wind speed, and incoming and outgoing longwave and shortwave radiation is used for the evaluation. When windspeed is only available below 10 m, as is the case for most of the glacier stations, the windspeed at 10 m is calculated using a logarithmic wind profile (assuming neutral stratification) with a roughness length of 1 mm. The assumption of neutral stratification, however, is a limitation, potentially having larger impact on the wind speed correction than sensor level alone. For Nordenskiöldbreen and Ulvebreen, measurements were conducted at ~4m above the surface, and the CARRA humidity and temperature is therefore interpolated to the measurement height by interpolation between the lowest model level (15 m) and 2 m.*

It is difficult to take into account the effect of snow accumulation, as many of the stations used do not measure this. We have instead added a section on the uncertainty introduced by adding 1 m of snow below the sensors, based on downscaling CARRA data and assuming a log-log profile of wind speeds:

*L187-200: The snowdepth is not measured at the majority of the used stations, and we therefore do not apply any correction factor due to changes in height after snow accumulation. The uncertainty associated with ignoring this effect depends on the specific variable (temperature, humidity, windspeed) and the measurement height. These uncertainties only affect the evaluation statistics, and not the model results.*

*Snow depths on Svalbard are modest and seldom amount to more than 1 m at most AWS sites. Assuming a snow depth of 1 m, a roughness length of snow of 1 mm, and that the windspeed can be approximated by a logarithmic profile (neutral stratification), the windspeed at 1 m above the surface is 7% lower than the windspeed at 2 m. For windspeeds measured at 10 m, decreasing the height by 1 m only amounts to a 1% decrease in windspeed. The windspeeds measured at the MET Norway stations and Kongsvegen, which are measured at 10 m, are therefore more robust to the effect of snow accumulation. The study by Østby et al. (2013) suggests a roughness length smaller than 1 mm which in turn would decrease the effect on wind speed.*

*It is trickier to estimate the uncertainties for temperature and relative humidity. Here, we use CARRA estimates of the temperature, pressure, wind speed, and humidity at the lowest model level (15 m) and at surface level (0 m) to interpolate the temperature and specific humidity, taking into account the stability of the atmosphere. The same method and parameters are used within CARRA to calculate variables at 2m height and is described in detail in the CARRA product user guide (Schyberg et al, 2020). The difference in temperature and humidity for all station locations is simulated for 2 m and 1 m above the surface over two different years (1994, a low melt year, and 2020, a high melt year). Even assuming the snowpack lasted the full year, the yearly average deviation was < 0.2◦C. The specific humidity at surface level was not available in CARRA, so for simplicity we assume fully saturated conditions. The yearly average difference in the results was always below 1%.*